# Value and choice as separable and stable representations in orbitofrontal cortex

Daniel L. Kimmel [1,2]✉, Gamaleldin F. Elsayed [3], John P. Cunningham [1,4] & William T. Newsome[5]

Value-based decision-making requires different variables—including offer value, choice, expected outcome, and recent history—at different times in the decision process. Orbitofrontal cortex (OFC) is implicated in value-based decision-making, but it is unclear how downstream circuits read out complex OFC responses into separate representations of the relevant variables to support distinct functions at specific times. We recorded from single OFC neurons while macaque monkeys made cost-benefit decisions. Using a novel analysis, we find separable neural dimensions that selectively represent the value, choice, and expected reward of the present and previous offers. The representations are generally stable during periods of behavioral relevance, then transition abruptly at key task events and between trials. Applying new statistical methods, we show that the sensitivity, specificity and stability of the representations are greater than expected from the population's low-level features—dimensionality and temporal smoothness—alone. The separability and stability suggest a mechanism—linear summation over static synaptic weights—by which downstream circuits can select for specific variables at specific times.

[1] Mortimer Zuckerman Mind Brain Behavior Institute, Columbia University, New York, NY 10027, USA. [2] Department of Psychiatry, Columbia University, New York, NY 10032, USA. [3] Google Research, Brain Team, Mountain View, CA 94043, USA. [4] Department of Statistics, Columbia University, New York, NY 10027, USA. [5] Department of Neurobiology and Wu Tsai Neurosciences Institute, Stanford University, Stanford, CA 94305, USA. ✉email: dkimmel@columbia.edu

For value-based decision-making, economists, psychologists, and neuroscientists have long-posited a need for separable representations of key decision variables, including (1) initial valuation of the offer, (2) chosen action, and (3) expected outcome given the choice[1–4]. These representations in turn support respective steps in the decision process: (1) committing to an initial behavioral policy, (2) computing expected outcome while executing the policy and then later assigning credit to the chosen action, and (3) re-evaluating the policy mid-execution and then later comparing expected and received outcomes to update future expectations.

However, study of the neural basis of this process has faced five significant limitations. First, reports differ on how neural populations represent multiple decision variables so as to drive distinct functions. Some studies suggest that single neurons, particularly in orbitofrontal cortex (OFC), specialize in representing single task-relevant variables (i.e., categorical selectivity)[5–8]. In contrast, recordings from many cortical areas—including OFC, dorsolateral prefrontal (PFC), anterior cingulate (ACC), and posterior cingulate cortices—indicate that single neurons encode multiple decision variables (i.e., mixed selectivity)[9–13], an encoding strategy that requires a means to de-mix these signals if they are to drive downstream functions independently.

Second, the task-relevant variable(s) encoded by a single neuron or population in OFC and other cortical areas often change over the course of the decision process[14,15]. Thus, a readout selective for one variable at one point in time, may in fact select for an entirely different variable 100's of milliseconds later. How can a single population support multiple downstream functions at distinct times[16]?

Third, existing analyses of OFC frequently obscure any mixed or dynamic selectivity. For instance, most studies report the percentage of neurons representing variables $X$ or $Y$ at each time point independently. This approach both implies categorical selectivity—neurons represent either $X$ or $Y$—and obscures how the contribution of individual neurons changes between time points. That is, even if the percentage remained constant, were the same neurons contributing at all time points, and if so, to the same extent? The implications for readout are substantial.

Fourth, though population-level analyses offer a means to de-mix single-neuron responses into separate representations of each variable[17–20], these analyses have not been applied to OFC, nor, in general, been coupled with a statistical framework to distinguish highly sensitive and/or stable representations from epiphenomena that often arise from low-level features of neural population activity[21].

Fifth, value signals are likely important at different phases of the decision process, and yet traditional value-based tasks test only some of these roles. In most studies, for example, subjects render choices with brief, all-or-nothing responses (e.g., a reach, lever press, nose poke, or saccadic eye movement). In these tasks, because the cost of responding is low, value informs which choice is rendered, but not whether a choice is rendered; indeed, subjects will almost always render a choice. In contrast, most ethological decisions require an agent to execute a behavioral policy over time, often with sustained effort (e.g., deciding to forage from a particular fruit tree, then sustaining that policy while competing with other animals)[22]. To apply the policy adaptively—exerting variable effort or even reversing the policy midway—requires a sustained neural representation of the expected value, which may not be elicited by the brief, all-or-nothing choices of most tasks.

Here, we present a novel behavioral task in which macaque monkeys trade sustained effort for juice rewards. Across trials, animals are more likely to accept larger offers, but within trials, their choices are not absolute: animals continue to re-evaluate, and at times reverse, their initial choice mid-trial. We simultaneously recorded from single neurons in macaque OFC and find heterogeneous responses exhibiting mixed and dynamic selectivity. To capture this complexity, we apply a new dimensionality reduction technique and a new statistical model that discovered separable population representations of the key task-relevant variables—offer size, choice, expected reward—for which the magnitude and specificity exceed that expected by chance from the population's low-level features. Moreover, the representations are stable during the task periods when the variables are behaviorally relevant, then change abruptly. Likewise, between trials, task-relevant information transfers rapidly to a new set of dimensions, thus maintaining previous-trial representations while distinguishing them from present-trial inputs. The dynamics of the representations—abrupt transitions at key task events followed by stability during periods of behavioral relevance—suggest that OFC organizes dynamically to represent task-relevant information at specific times. The low-dimensional, stable nature of the representations suggests a neurobiologically plausible mechanism—linear summation over static synaptic weights—by which downstream circuits can read out mixed, heterogeneous responses into separable representations for driving specific behavioral functions at specific times.

## Results

**Subjective value increases smoothly with increasing benefit**. In a novel cost-benefit task, two macaque monkeys, N and K, decided whether to trade sustained effort for juice reward. The number of juice drops offered was presented briefly as 0, 1, 2, 4, or 8 yellow icons and varied randomly across trials (Fig. 1a). To accept an offer, the animal maintained visual fixation for a constant duration (work period)—an effortful process with economic cost[23–25]—and then received the promised reward. To reject an offer, the animal averted its gaze and waited for the next trial. We analyzed 9637 and 27,952 trials from 26 and 86 experimental sessions (monkeys N and K, respectively).

As expected, the likelihood of the animal accepting an offer increased smoothly with offer size (on average, Fig. 1b; in individual sessions, Supplementary Fig. 1) and did not depend on the offer's visual properties (singleton offer; Fig. 1b, open circles). This suggested that, on each trial, animals computed the value of the offer based on its variable benefit (drops of juice) and fixed cost (effort).

**Mid-trial re-evaluation of choice**. A key task feature was the sustained effort required to accept an offer, allowing the animal to accept an offer initially, but continuously re-evaluate its choice and, at times, change its mind, rejecting the offer mid-trial. We leveraged the timing of fixation breaks—measured as the rejection hazard rate (Fig. 1c; Supplementary Fig. 2)—to infer the dynamics of the underlying decision process. (Here we focus on the non-zero offers, which were qualitatively distinct from 0-reward offers; see Supplementary Fig. 3.) For a given offer size, the hazard rate peaked within 1–1.5 s of the offer (early phase), suggesting most decisions occurred early in the trial, as the task structure incentivized.

However, a second phase of rejections occurred mid-trial—from ~1 s after the offer until the final 1–2 s of the work period (mid-trial phase)—when the animal began the period of sustained effort and the external offer cue was removed. Intriguingly, the hazard rates during the mid-trial phase were qualitatively distinct—lower magnitude and slower dynamics—from the early phase, though they continued to depend on offer size (implying a memory of the offer) and were progressively decreasing (consistent with volitional rejections and not accidental breaks in fixation—see late phase, below). These features suggested a

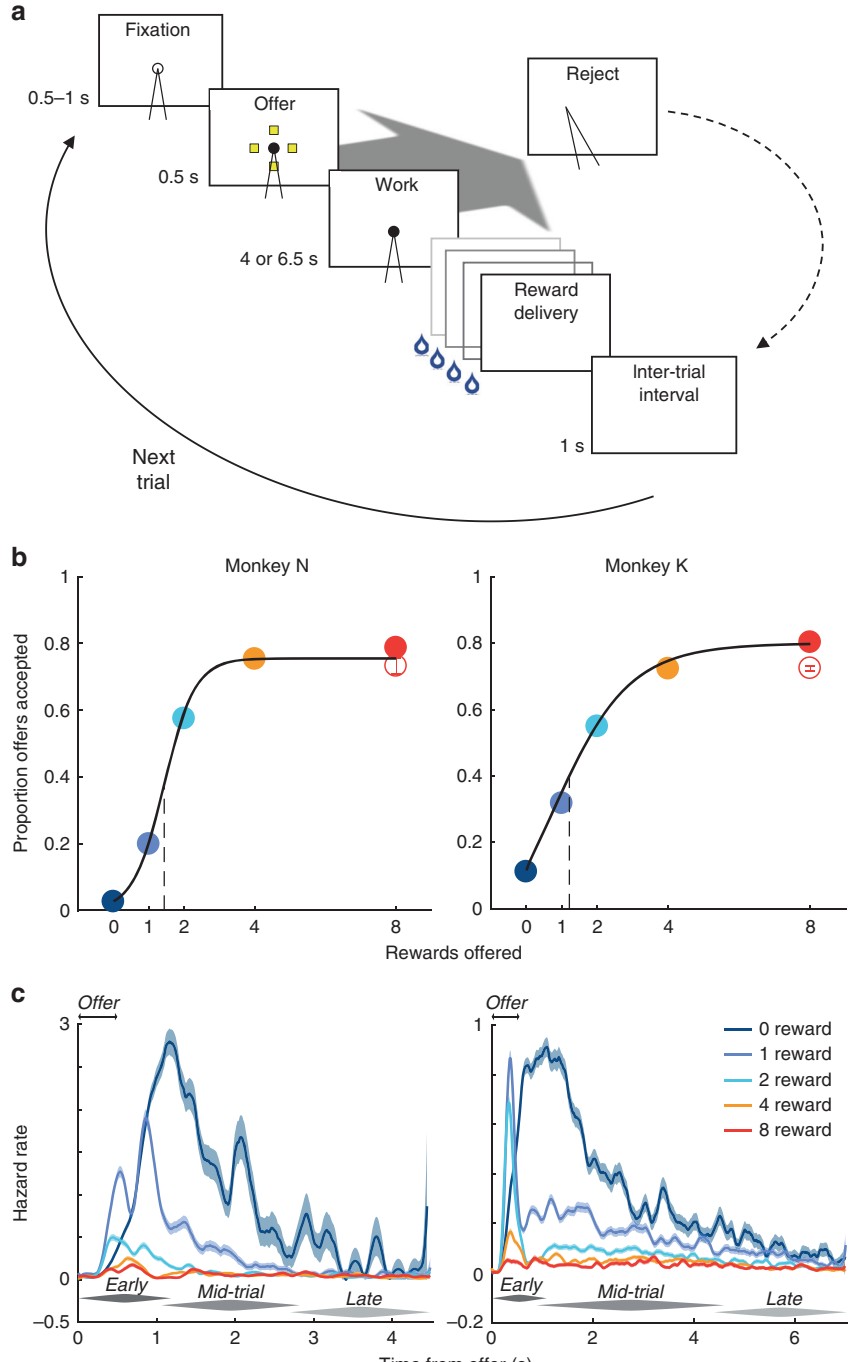

**Fig. 1 Behavioral design and results. a** The time course of a trial is shown proceeding from top-left to bottom-right. The animal initiated the trial by directing its gaze toward an open circle, or fixation point (FP), and held fixation for a variable interval. An offer was presented as an array of icons (0, 1, 2, 4, or 8 yellow icons, selected randomly across trials) with each icon indicating a drop of juice in exchange for effortful work; simultaneously, the FP filled-in to a solid circle (providing a timing cue on 0-reward offers). To accept the offer, the animal sustained fixation through the work period (4 or 6.5 s, monkey N or K, respectively), after which the promised drops of juice were delivered in rapid succession. To reject the offer (large oblique gray arrow), the animal averted its gaze during the offer or work period. The trial then entered a timeout period equivalent in duration to the time that would have elapsed had the animal accepted the offer. The trial was followed by an inter-trial interval before advancing to the next trial (solid curve). **b** The proportion of offers accepted is shown as a function of offer size, which was conveyed to the animal either by the number of yellow icons (non-singleton offers; filled circles) or as single purple icon worth 8 drops (singleton offer; open circle), which controlled for the inherent confound between reward magnitude and the visual impact of more icons. Data are presented as proportion (circles) ± s.d. of the proportion (error bars; generally smaller than circle) from $N = 9637$ or 27,952 trials from monkeys N or K (trial counts evenly distributed across offer sizes). Logistic function was fit to aggregated data for display purposes only (black curve); all statistical tests in Supplementary Fig. 1 were performed on individual sessions. **c** Hazard rate (colored curves) ± s.d. (shading) of fixation breaks are shown for non-singleton offers as function of time from onset of the offer period (double arrows). Early, mid-trial, and late phases of rejections referred to in main text are indicated by dark, middle, and light gray tapered bars, respectively.

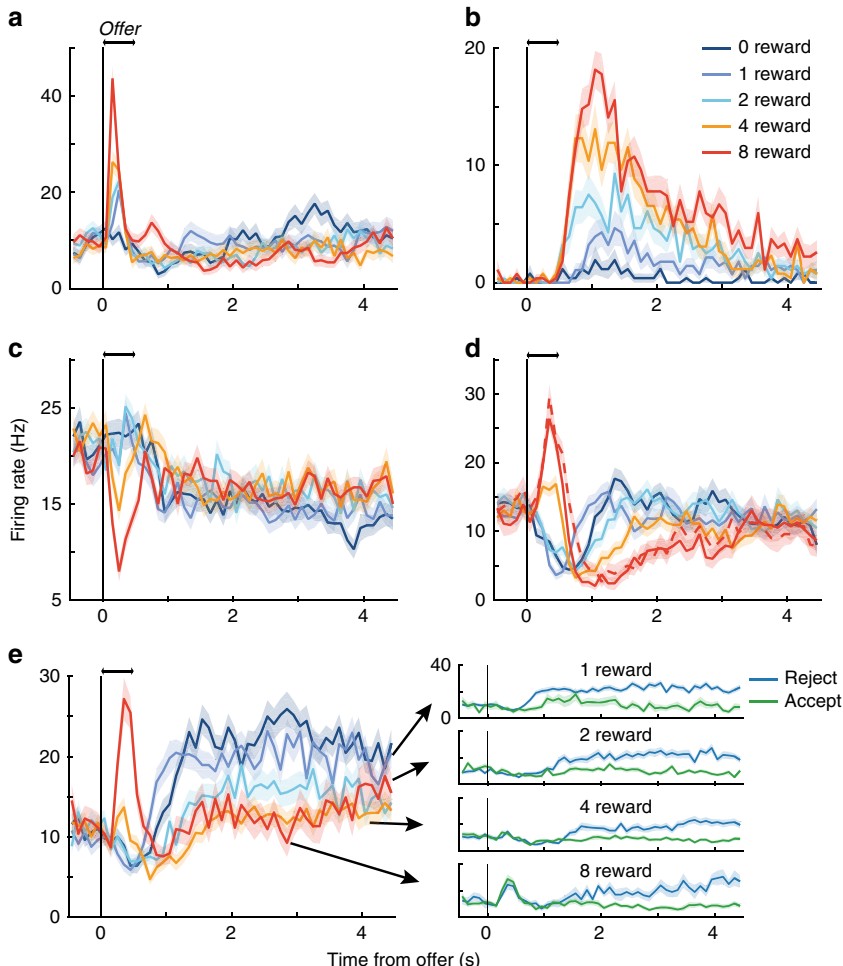

**Fig. 2 Dynamic encoding and mixed selectivity in individual units. a–e** Mean response to the non-singleton (solid curves) and singleton (dashed curve; not present in all sessions) offers ± s.e.m. (shading) is shown as a function of time from onset of the offer period (double arrows) for five example units. Averages include all choices. Dynamics varied from rapid and transient (**a**, **c**) to slow and sustained (**b**). The sign of encoding could be positive (**a**, **b**) or negative (i.e., firing less for larger offers (**c**)), or reversing mid-trial (**d**). Further stratifying by accept (green) and reject (blue) choices (**e**) showed selectivity for both benefit and choice. While the encoding of offer size appeared to reverse sign for unit (**e**), this was an artifact of the greater response for reject than accept choices later in the trial combined with more frequent rejections for smaller offers. In contrast, the reversal observed for unit (**d**) did not depend on choice (not shown).

separate decision process: on a small fraction of trials, the animal initially accepted the offer, but later changed its mind and rejected the offer mid-trial[26,27]. Several value-based variables likely evolved over the trial and may have contributed to these mid-trial reversals, such as an increase (decrease) in expected benefit (cost) as the time-to-reward shortened[23]. Presumably, these mid-trial choices required a sustained neural representation of value, possibly distinct from the representation supporting the early-phase choices, as we explored below.

Finally, for the last 1–2 s of the work period (late phase), the hazard rate for most non-zero offers converged to a low and static level that did not depend on offer size. A static hazard rate (i.e., events occurring independently over time) is a feature of a Poisson process[28], consistent with accidental failures of fixation (i.e., "falling off the log"), and may partly account for the maximal accept rates of <100%.

**Heterogeneous encoding of task variables in individual units.**
We analyzed the spiking activity of 68 and 342 units from OFC in monkeys N and K, respectively (see "Methods"; Supplementary Fig. 4). Responses from single units and multi-units did not differ qualitatively and were analyzed collectively as individual units

(see Supplementary Figs. 30–32 for analysis of single units exclusively).

As expected, individual units were robustly modulated by offer size, consistent with encoding stimulus value. However, we observed marked heterogeneity in the dynamics and sign of encoding (Fig. 2a–d). Moreover, individual units were sensitive to multiple variables, e.g., encoding not only offer size, but also the animal's choice Fig. 2e. Conventional approaches for summarizing this heterogeneity (e.g., classifying responses into discrete categories and reporting the percentage of units per category and time bin[5,15]), obscured how the encoding strength, sign, and/or selectivity of an individual unit changed over the trial—dynamics that were clearly evident in individual responses.

**Discovering low-dimensional representations of task variables.**
Dynamic encoding and mixed selectivity presented a challenge both for describing the population activity and for considering how downstream circuits may decode, or read out, a specific variable at a specific time. Building on prior work[18], we developed a new analysis, optimal targeted dimensionality reduction (oTDR; see "Methods"), that discovered and quantified population-level representations. In brief, we identified a priori the behavioral

variables relevant to performing the task: benefit (i.e., number of rewards offered), choice (i.e., accept or reject), and expected reward (i.e., benefit × choice, which reflected the outcome given the choice). We quantified how strongly each unit encoded the task-relevant variables by the coefficients derived from linear regression of trial-average firing rate (within each condition) on the task-relevant variables. Across the population of $N$ units, the regression coefficients for a given variable defined an $N$-dimensional vector, i.e., regression axis (RA) or population representation, which best linearly represented the variable.

We explored two classes of RAs: (1) dynamic RAs (dRAs) that were calculated for each time bin and tested how representations changed over the trial, as discussed below, and (2) static RAs (sRAs) that tested the suitability of a fixed readout for representing the task-relevant variables throughout the trial. The sRA for a given variable was calculated in the temporal epoch in which we hypothesized a priori that the variable was behaviorally relevant. Specifically, we reasoned that the animal encoded external value information during the offer period (0–0.5 s), and thus computed the BENEFIT sRA in this epoch. After the offer, the decision process proceeded without external information and thus depended on internal representations of the task variables—CHOICE and EXPECTED REWARD—which were computed during the post-offer, work period (0.5 s–…). We confirmed that our conclusions below were robust to a range of temporal epochs (Supplementary Fig. 12). (We refer to the names of sRAs in small-caps and to task-relevant variables in lowercase, e.g., BENEFIT refers to the sRA representing the benefit variable.)

**Mixed selectivity for task variables in individual units**. We examined the sRA coefficients to ask whether individual units were indeed selective for multiple variables (i.e., mixed selectivity), as in Fig. 2e, or instead specialized for single variables (i.e., categorical selectivity)[5,29]. If specialized, then for a given variable, the absolute value of the corresponding sRA coefficient would be large for some units (i.e., highly selective), but concentrated near 0 for most units (i.e., non-selective), resulting in a heavy-tailed distribution. However, the distributions of coefficients were not significantly different from Gaussian (Supplementary Fig. 5a, b), consistent with random assignment of coefficients to units, independently of the other variables (i.e., not specialized). Likewise, we did not observe an anatomical organization (e.g., clusters or gradients) of sRA coefficients (Supplementary Fig. 6).

Across variables, we found that an individual unit significantly encoded two or more variables at a frequency that was (a) much greater than expected if units were specialized for a single variable and (b) statistically consistent with independent selectivity across variables (Supplementary Table 1). In addition, the magnitude of encoding between pairs of variables did not favor a single variable and was sufficiently reliable to exclude noisy coefficients masquerading as mixed selectivity (Fig. 3; Supplementary Fig. 7a, c; Supplementary Table 2, "Within-variable" section). In summary, we observed no statistical evidence that individual units were preferentially selective for single task-relevant variables.

In a separate analysis, we confirmed that mixed selectivity arose in single units, and was not, for example, an artifact of pooling categorical single-unit responses (Supplementary Fig. 30; Supplementary Table 1).

**Separability of low-dimensional representations**. Though a given unit may encode two or more variables, if the extent of encoding were sufficiently correlated across units, then the population representations (i.e., sRAs) would not be separable by a downstream observer. Pairs of sRAs were at most modestly correlated (Fig. 3b, d, open bars; Supplementary Table 2). However, conventional statistics addressed the chance of falsely concluding orthogonal representations were correlated, whereas we were interested in the opposite extreme: whether two (possibly correlated) representations carried (at least some) independent information (see "Methods"). Indeed, the representations were highly separable—i.e., correlations between sRAs were significantly less than expected for perfectly correlated representations corrupted by independent noise (Fig. 3b, d, open bars vs. dashed lines; Supplementary Fig. 7b, d and Supplementary Table 2).

**Reading out activity of low-dimensional representations**. Though selectivity was mixed in individual units, oTDR implied a neurobiologically plausible mechanism by which a downstream population could de-mix these signals into representations specific to each variable simply by tuning synaptic weights to the sRA coefficients. We computed this weighted sum, or activity of each representation, by projecting the high-dimensional trial-average firing rate onto each sRA. To ensure independence between the projections (and thus permit our statistical assessments), we constrained the sRAs to be orthogonal for this and subsequent sRA analyses. (The analyses in the preceding two sections used the unconstrained relationships between the sRAs.)

Examining sRA activity (Fig. 4a, b), BENEFIT discriminated offer size rapidly and robustly after presentation, but the selectivity decayed abruptly and was absent by 1.5 s. Moreover, the representation did not discriminate the animal's choice (thick and thin curves overlap). In contrast, CHOICE discriminated accept and reject choices beginning around 2 s for most offers and with increasing selectivity through the trial (see Supplementary Fig. 8 for consideration of single-trial dynamics and post-rejection gaze), but did not discriminate offer size. Thus BENEFIT and CHOICE de-mixed information about their respective variables.

To more directly test the link between sRA dynamics and choice timing, we compared CHOICE activity for early vs. late rejections and found that choice selectivity emerged later on late-rejection trials, consistent with the representation reflecting the underlying decision dynamics (Supplementary Fig. 9). However, CHOICE activity did not discriminate choice until after the median rejection time (0.92 or 1.3 s, monkey N or K) and after choice information was integrated into the EXPECTED REWARD representation (see below). Thus CHOICE likely reflected, rather than predicted, the decision. Moreover, at the single-trial level, we found no evidence that individual OFC responses to the current offer predicted the upcoming choice (i.e., choice probability; Supplementary Fig. 11).

Finally, EXPECTED REWARD discriminated offer size as early as 500 ms—as offer information transitioned abruptly from the BENEFIT sRA—but did so only for accept choices. As such, EXPECTED REWARD integrated benefit and choice: activity reflected offer size for accept choices, but was undifferentiated for rejected offers. Temporally, EXPECTED REWARD coincided with the period of mid-trial rejections when no stimulus was present and thus may have provided the internal value representation necessary for mid-trial re-evaluation (Fig. 1c).

**Validity, specificity, and sensitivity of representations**. For a static, low-dimensional representation (i.e., sRA) to serve as a suitable read out, the variance it explains should be greater than alternative dimensions (validity), unrelated to the other variables (specificity), and capture all of the available relevant variance, obviating the need for additional dimensions (sensitivity). To test these criteria, we defined relevant signal variance (RSV) as the portion of variance explained that was linearly related to the

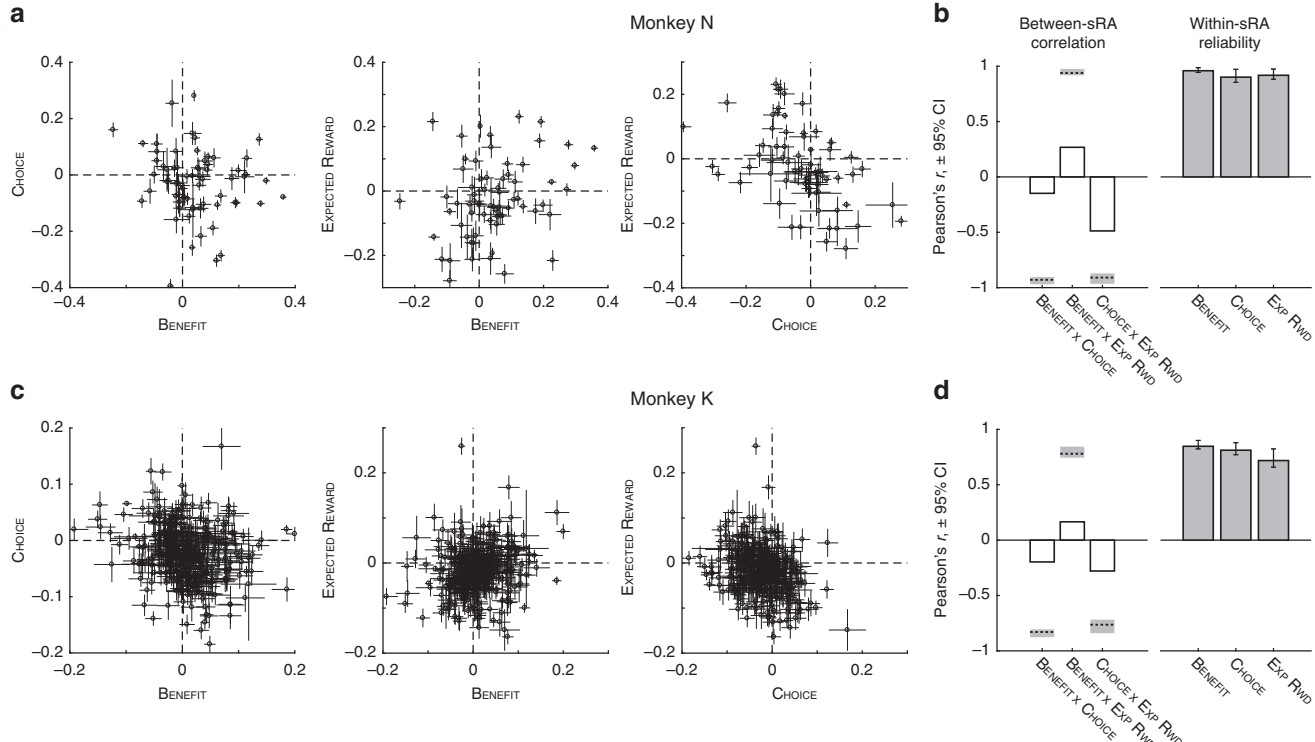

**Fig. 3 Contribution of individual units to low-dimensional representations. a**, **c** The pairwise relationship between regression coefficients for task-relevant variables (abscissa and ordinate labels) are shown for each unit (open circles; point estimates derived from all trials) with standard deviations (horizontal and vertical error bars; computed by resampling trial-level responses, $N = 700$ resampled data sets per unit, see "Methods"), for monkeys N (**a**) and K (**c**). If units generally encoded a solitary variable, circles would cluster on the horizontal and vertical meridians (dashed lines), which was not observed, consistent with mixed selectivity. If estimates of the regression coefficients were unreliable, then apparent mixed selectivity could arise spuriously from imprecise estimates and error bars would consistently overlap the meridians, which was not observed. **b**, **d** The Pearson's correlation coefficient between pairs of representations (i.e., sRAs) of different (open bars) or same (filled bars) variable(s) are shown for monkeys N (**b**) and K (**d**). For between-sRA correlations, the coefficient (bar height) was measured from the full data set; dashed horizontal line and shading indicate the mean and 95% confidence interval (CI), respectively, of the hypothetical correlation between two perfectly correlated (or anti-correlated) representations corrupted by independent noise (i.e., low precision of the individual-unit coefficients). Observed correlations were significantly closer to zero than these hypothetical values, defining the representations as separable. For within-sRA correlations, the coefficient mean (bar height) and 95% CI (error bars) were measured from the resampled data sets. The resampled data for both between- and within-sRA analyses included 700 resampled data sets generating $N = 244{,}650$ pairwise correlations. See full distributions in Supplementary Fig. 7 and summary statistics in Supplementary Table 2. For present figure, all regression coefficients pertain to non-orthogonalized sRAs, which most closely reflected the population encoding and permitted meaningful comparison between representations (in contrast, the correlation between orthogonalized sRAs would necessarily be 0).

sRA's targeted variable (Fig. 4c, d, solid curves; see Supplementary Fig. 13 for total variance explained).

Indeed, the RSV for each sRA was significantly greater than chance during its period of behavioral relevance (Fig. 4e, f): encoding external value information (BENEFIT), sustaining value information during the work period (EXPECTED REWARD), and representing the chosen action (CHOICE). Critically, we determined chance levels from a set of random vectors whose density reflected the data's dimensionality, producing more conservative (i.e., larger) p-values than evenly distributed vectors, as used typically (see "Methods").

The sRAs were highly specific to their targeted variables, explaining generally low and non-significant levels of residual, or irrelevant, signal variance (ISV; Fig. 4c–f, dashed curves). Moreover, this residual variance had little relation to the other task variables, which would otherwise confuse a downstream observer (Supplementary Fig. 14).

The sRAs were sensitive to their targeted variables for some, but not all, times in the trial. We considered two explanations: (1) the variables were simply not represented in any dimension during these times, and/or (2) the dimensions representing the

variables rotated (i.e., the contribution of individual units changed) during the trial, aligning with the sRAs for only a portion of time. To test these hypotheses, we computed the optimal representations independently in each time bin (i.e., dRAs). At a given time, the dRA magnitude (prior to normalization) defined the extent of population encoding, whereas the angle $\theta_{ij}$ between the dRAs at times $i$ and $j$ measured the rotation of the encoding dimension.

We found qualitative evidence for both hypotheses: (1) the available signal, in any dimension, fluctuated over the trial (Fig. 5a, b); and (2) the dimension representing a given variable was consistent for discrete periods (Fig. 5c, d, areas of warm colors), but for some variables, changed abruptly mid-trial, producing multiple, distinct representations (e.g., benefit representation early vs. mid-trial, highlighted with brackets in left panels of Fig. 5e, f), thus limiting the generalizability of any static dimension (i.e., sRA). Nonetheless, the available signal uncaptured by the sRAs (i.e., "left on the table") was generally small and never greater than expected for a random set of three static dimensions (Supplementary Fig. 15).

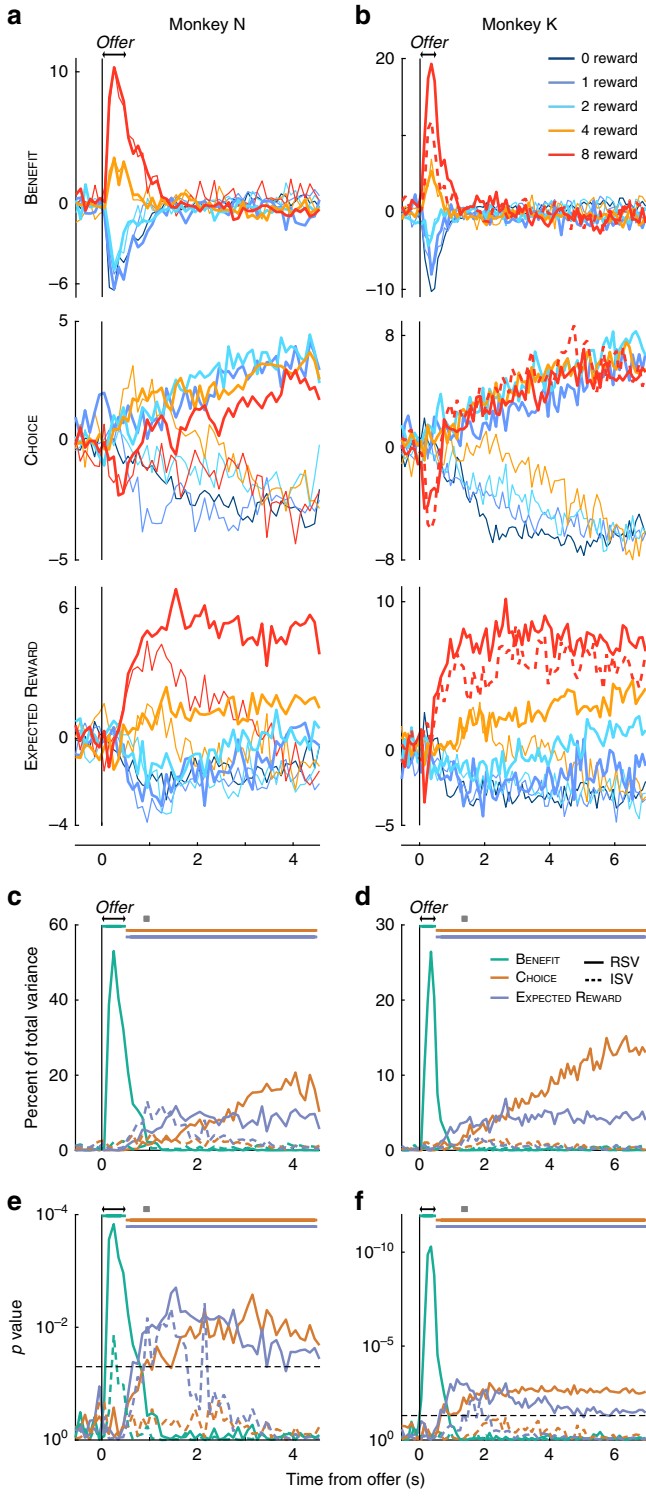

**Fig. 4 Activity of and variance explained by low-dimensional representations. a, b** The time course of the high-dimensional neural activity projected onto the low-dimensional representations (i.e., activity of the sRA) of BENEFIT (top), CHOICE (middle), and EXPECTED REWARD (bottom) is shown in arbitrary units for each offer size (colors) and accept and reject choices (thick and thin curves, respectively) as a function of time from the onset of the offer period (double arrows) for monkeys N (**a**) and K (**b**). The response to the singleton condition (dashed curve), which was excluded when computing the sRAs, tracked the offer's value, not visual properties (see Supplementary Fig. 10 for monkey N). Note that some combinations of offer and choice (e.g., 0-reward, accept choices) had too few trials per unit to accurately estimate trial-average responses and were excluded (see "Methods"). The apparent bleed-through of offer information onto mid-trial CHOICE activity for reject choices (middle panel, thin lines), was in part because rejections (and thus choice selectivity) occurred later for larger offers (Supplementary Figs. 2 and 8), and in part because the time-varying representations (i.e., dRAs) were correlated mid-trial (Supplementary Fig. 24, left column), despite orthogonal static representations. **c, d** The relevant and irrelevant signal variance (solid and dashed curves, respectively) for BENEFIT (green), CHOICE (orange), and EXPECTED REWARD (blue) are shown as a percentage of total cross-condition variance as a function of time from the onset of the offer period (double arrows) for monkeys N (**c**) and K (**d**). **e, f** $\log_{10}$ probability (one-sided, uncorrected) of data in **c** and **d**, respectively, was derived empirically in comparison to random dimensions (see "Methods"). Horizontal dashed line corresponds to $p = 0.05$. In **c–f**, gray squares indicate the median rejection time and colored horizontal bars span the temporal epoch in which the color-matched sRAs were computed.

temporal span of each boxcar defined a period of putative stability, and the boxcar height measured the magnitude of similarity within that period (Supplementary Fig. 21; note: high similarity corresponded to small angles $\theta_{ij}$).

We next developed a statistical framework to assess the significance of these periods. For instance, how surprising was it that the choice representation changed by only 20° in a 3 s interval? To address such questions, we computed the similarity during identical periods (i.e., boxcar spans from the veridical data) in synthetic firing-rate data. Critically, the synthetic data did not encode the task variables but preserved the veridical data's dimensionality and temporal smoothness—low-level features that bounded how much a representation could change, and thus could generate similarity trivially (see "Methods" for details and intuition). A period was statistically stable when the observed similarity was significantly greater than the distribution of null similarity across synthetic data sets (Supplementary Fig. 21c, f).

Most representations in OFC were statistically stable during discrete periods when the encoded information was behaviorally relevant (Fig. 5e, f, colored horizontal bars). Specifically, the representation of benefit was stable for a period of ~0.5 s (Fig. 5e, f, left panel, red bracket) coinciding with the offer presentation. Subsequently, a new representation of benefit emerged (blue bracket) that, while stable for ~3 s, was dissimilar from the earlier period and weaker in magnitude (Fig. 5a, b). The representation of choice was stable from ~1.5 s (after most choices were rendered) to the end of the trial. The representation of expected reward after ~1 s (around the median rejection time) was stable for monkey N but equivocal for monkey K (see Fig. 5 legend).

In summary, the low-dimensional representations of the task-relevant variables were statistically stable, i.e., the specific contributions of individual units were more consistent than expected by the data's low-level features. And yet the timing of these stable periods varied across the task variables, aligning with the concurrent task demands for which the representation was behaviorally relevant.

**Stability and task-alignment of dynamic representations.** Thus far, we observed that the dynamic representations (dRAs) were qualitatively similar for discrete periods of the trial. To formalize these impressions, we systematically delineated periods of similarity and then tested these periods for statistical stability—i.e., unusually high or protracted similarity—which suggested a representation may have a specialized functional role (see "Methods"). Briefly, we fit boxcar functions to each row of the heat maps in Fig. 5c, d (see Supplementary Fig. 20 for fits). The

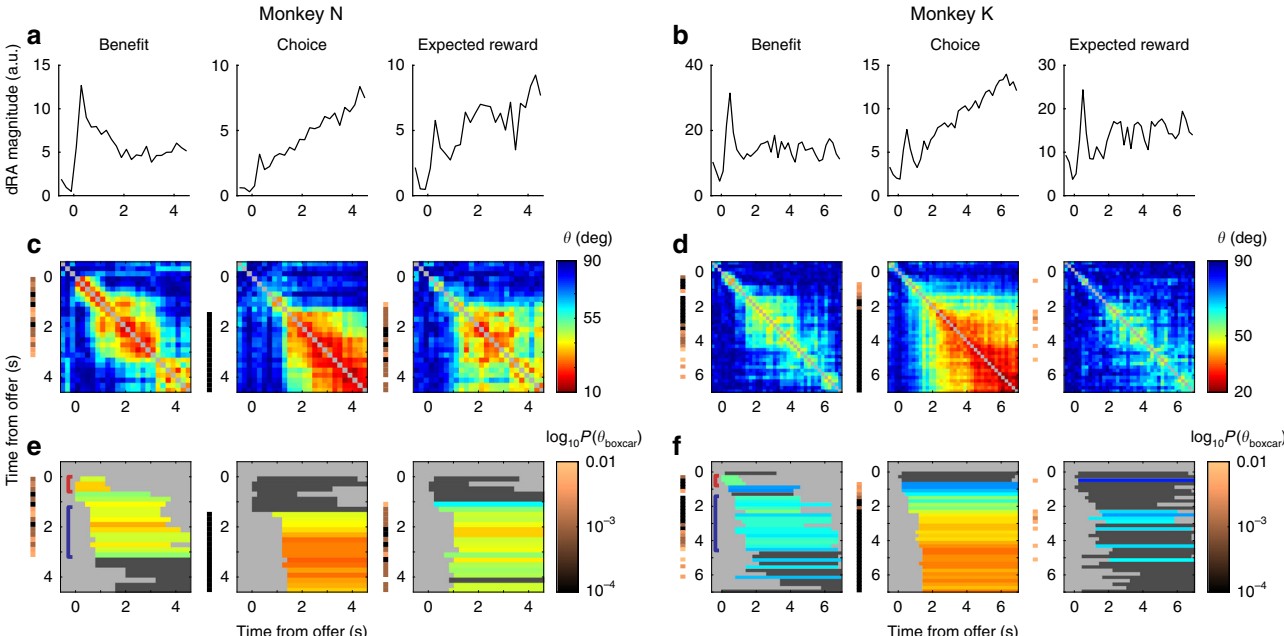

**Fig. 5 Stability of dynamic representations.** Features of the dynamic low-dimensional representations (dRAs) of benefit, choice, and expected reward (left, middle, and right columns, respectively) are shown for monkeys N (**a**, **c**, **e**) and K (**b**, **d**, **f**). **a**, **b** The vector magnitude of each dRA, aligned to the offer period, indicates the time-varying extent of population encoding, which was useful for interpretation (e.g., high similarity may be more meaningful when between more robust, higher magnitude dRAs). **c**, **d** Angle $\theta$ in degrees between pairs of dRAs computed at different times relative to the offer period (given by row and column positions) is shown by the color of each pixel (referencing right-hand color scale in **c** and **d**). Smaller angles (warmer colors) correspond to greater similarity between representations. The diagonal compares identical dRAs ($\theta = 0°$) and is colored gray. **e**, **f** A boxcar function was fit to each row of angles in **c** and **d**, with the boxcar height ($\theta_{boxcar}$) indicating the average similarity. The period of non-zero boxcar height is shown by a colored or dark gray horizontal bar in the corresponding row of **e** and **f**. For periods of significant similarity (i.e., $p(\theta_{boxcar}) < 0.01$), bar color indicates boxcar height, referencing the right-hand color scale in **c** and **d**. (For visual consistency, boxcar height was inverted such that higher boxcars, i.e., greater similarity, corresponded to smaller angles.) In contrast, periods of non-significant similarity are shaded dark gray. This distinction can be seen in the expected reward representation for monkey K: periods of extended similarity were observed during the work period, but these periods only occasionally achieved statistical stability (see $p(\theta_{boxcar})$, Supplementary Fig. 21f). The thin vertical bands to the left of each panel in **c**–**f** show the $\log_{10}$ probability (one-sided; uncorrected; derived empirically in comparison to synthetic data, see "Methods") of observing $\theta_{boxcar}$ for the corresponding row, with colors referencing the right-hand color scale in **e** and **f**; no color is shown when $p(\theta_{boxcar}) \geq 0.01$. Colored brackets in left-most panel highlight specific periods referenced in main text. Our conclusions were robust to the choice of time bins (Supplementary Fig. 22). For statistics on the pairwise angles $\theta_{ij}$, see Supplementary Fig. 19.

**Representations transition abruptly between trials.** Given the magnitude of the static representations (Fig. 4c, d) and stability of the dynamic representations (Fig. 5c, d) during the trial, we asked whether task-relevant information was preserved across trials. Although not required by our task design, the animals nonetheless may have used choice or reward history from the previous trial to inform upcoming decisions, a phenomenon observed even in tasks that do not incentivize cross-trial integration[5,30].

To observe the transition between trials, we defined a new analysis window aligned to fixation on the present trial and extending retrospectively into the previous trial. Intuitively, we expected the static representations (sRAs) would encode information about the coincident trial, regardless of whether it was designated "present" or "previous." Indeed, during the previous trial, the sRAs explained variance related to the variables on the previous trial (Fig. 6a–d, left panels), just as they had explained variance related to present-trial variables during the present trial (Figs. 4c–f and 6a–d, right panels). (To assess previous-trial encoding, we recomputed trial-average responses according to the previous-trial variables: previous benefit, previous choice, and experienced reward; see "Methods" for details and rationale.)

Between trials, the CHOICE sRA maintained information about the preceding choice into the ITI (Fig. 6a–d, orange curves). However, just before fixation on the new trial, information about the previous trial disappeared precipitously (Fig. 6a–d, left panels, orange and, to lesser extent, blue curves as approaching time = 0), as though the system were clearing its cache to accommodate information about the upcoming trial. At first blush, there appeared to be no continuous representation of the task-relevant variables across trials. Yet, we wondered whether previous-trial signals were transmitted in other dimensions, outside the subspace defined by the sRAs. Using oTDR, we searched deliberately for such dimensions by computing previous-trial sRAs (PREVIOUS BENEFIT, PREVIOUS CHOICE, and EXPERIENCED REWARD): representations of the previous-trial variables during the first 0.5 s of fixation on the new trial, a time when the animal was under behavioral control but not yet exposed to the new offer.

Remarkably, just before the animal fixated on the new trial, the previous-trial sRAs suddenly and robustly represented the events from the previous trial, filling-in the temporal gap left by the present-trial sRAs (Fig. 6e–h vs. a–d), as though information were passed between distinct sets of neural dimensions (confirmed below). Extending into the new trial (Fig. 6e–h, right panels), PREVIOUS CHOICE (orange curves) continued to represent the previous decision through the new offer and until the time of most rejections (~1 s). In addition, in monkey K, PREVIOUS OFFER and EXPERIENCED REWARD significantly encoded their respective variables through all or part of this pre-rejection period (green and blue curves, respectively). As such, the timing of the

previous-trial representations was sufficient to influence the new choice based on the previous trial's outcome.

Remarkably, the previous-trial sRAs explained variance only after the trial had ended, but not during the previous trial itself, which was the domain of the present-trial sRAs (Fig. 6, left panels, time ≪ 0, Fig. 6e–h vs. a–d). This implied that the present- and previous-trial sRAs spanned separate subspaces. Indeed, they were at most weakly correlated and corresponded to highly dissimilar, nearly orthogonal dimensions (Supplementary Fig. 25; Supplementary Table 3). That is, OFC maintained distinct representations not only of offer, choice and reward events, but also of the relative trial in which those events occurred, allowing a downstream readout to distinguish previous-trial information (e.g., PREVIOUS CHOICE) from its present-trial counterpart (e.g., CHOICE).

## Discussion

We found that single neurons in macaque OFC represented key task-relevant variables—benefit, choice, and expected reward for both the present and previous trials—while the animals made cost-benefit decisions requiring sustained effort. For individual neurons, the encoding of value and choice was mixed, and the time course of encoding varied widely across neurons. However, using a set of new analysis and statistical techniques, we de-mixed the task-relevant signals into static low-dimensional representations that were separable at the level of the population. In addition, the time series of dynamic representations were statistically stable during periods when the information was behaviorally relevant, and then transformed abruptly at key task events and between trials, as information transitioned between dimensions. Our findings suggest that OFC reorganizes—forming and disbanding coordinated combinations of neurons—on behavioral timescales to represent and manipulate information relevant to concurrent behavioral demands.

Prior reports have argued for categorical selectivity in OFC— i.e., individual units specialized to encode a single variable—either implicitly by classifying otherwise mixed responses into single-variable categories, or explicitly by comparing univariate regressions performed independently for each variable[5–8,15]. In contrast, when variables competed for variance in a multivariate model, we found that units encoded two or more variables at rates equal to or above chance, consistent with broad evidence for mixed selectivity across cortical areas[9–13] and shown indirectly for OFC[29].

Moreover, we found the mixed representations were separable at the level of the population. That is, single neurons conveyed reliable, independent information about two or more variables, consistent with random assignment of encoding strength across neurons. Importantly, we distinguished the observed representations from mixed selectivity that may have appeared separable, but instead arose spuriously from unreliable estimates of otherwise perfectly correlated representations.

Several studies in rodent cortex have reported sequences of activity, where the neuron(s) selective for a given task variable changed on the order of 10's of milliseconds[14,31–33]. This unstable selectivity may be well-suited for representing variables that vary smoothly in time, such as spatial position, or for generating an eligibility trace for learning.

In contrast, the population representations in OFC were substantially stable, i.e., the selectivity of a given neuron was consistent during the behaviorally relevant period (see Supplementary Discussion on classifying stability). This permits a downstream circuit not only to select for the representation of a single variable, but to do so via a static set of synaptic weights (as prescribed by the sRA coefficients). Unlike the rapid, within-trial updating of readout weights implied by unstable selectivity, these weights could be tuned gradually during learning and then remain constant during mature behavior, a neurobiologically plausible mechanism[34].

The stability of the OFC population representations was not uniform across the trial or task variables, unlike prior reports of qualitative similarity for a single variable[13,35]. Rather, the representation of a given variable was stable for a period, then, at key task events, changed abruptly to a new, stable representation. This time-dependence could facilitate temporal gating—that is, not just which variable is read out, but also when that variable influences downstream computation. For instance, benefit was encoded along two, highly dissimilar dimensions during the offer and work periods (Fig. 5e, f, brackets). Therefore, a readout tuned to the earlier representation (which corresponded to the BENEFIT sRA) would be minimally sensitive to the later representation, and vice versa. This would permit the same information (i.e., benefit) to drive different functions. For instance, the early representation could inform the initial decision, while mid-trial benefit could integrate with choice to compute expected reward. A similar temporal gating could apply to choice information, with the CHOICE sRA informing credit assignment during the outcome, while PREVIOUS CHOICE biased the decision on the next trial. In motor cortex, analogous gating mechanisms may select for planning vs. execution signals[36,37].

Taken together, the features of OFC encoding—mixed but separable selectivity and stable population representations— would facilitate flexible readout of single variables, increase coding capacity, and implement non-linear operations between variables (e.g., EXPECTED REWARD)[10–12]. Importantly, these encoding features also provide a mechanism for temporal gating such that downstream circuits could tune synaptic strengths to selectively read out specific task-relevant variables during specific time periods.

These benefits of separable and stable selectivity satisfy the exigencies of a recent proposal that OFC encodes a cognitive map —a general, multipurpose framework for representing the set of cognitive and behavioral states relevant to the subject's current goals[38–40]. As behavioral demands change rapidly, so must the specific states represented by a cognitive map. This framework helps reconcile the multitude of functions proposed for OFC— and, in humans, ventromedial PFC—on the basis of seemingly disparate phenomenon[16,41]. Under a cognitive map, these diverse functions are unified as merely time-varying instantiations of a general process. Consistent with this model, within the same OFC neurons, the multiple representations needed to support many of OFC's proposed functions emerged dynamically according to concurrent behavioral demands, including encoding of external stimulus value (BENEFIT)[15,42–47], reward expectancy to support on-the-fly re-evaluation and prediction errors (EXPECTED REWARD)[48–51], and selected action to facilitate credit assignment (CHOICE)[52] and drive history-dependent biases (PREVIOUS CHOICE)[5,53].

Between trials, the remarkable transition of task information between nearly orthogonal representations suggested an intriguing neural implementation of temporal order, as though what we label as "present" vanishes into the "past" by entering a new dimension of neural space[54]. Perhaps our discrete concepts of past, present, and even future (i.e., prediction) are encoded not by separate neural populations or brain regions, but by separable dimensions within a single population.

The particular task design determines both which and when states must be represented[38]. Our novel task permitted animals to initially accept an offer and then, because the decision was rendered over time, to reverse their choice midstream. This behavior is akin to deciding to climb a banana tree or attend graduate school, choices which are still binary in nature, but can be

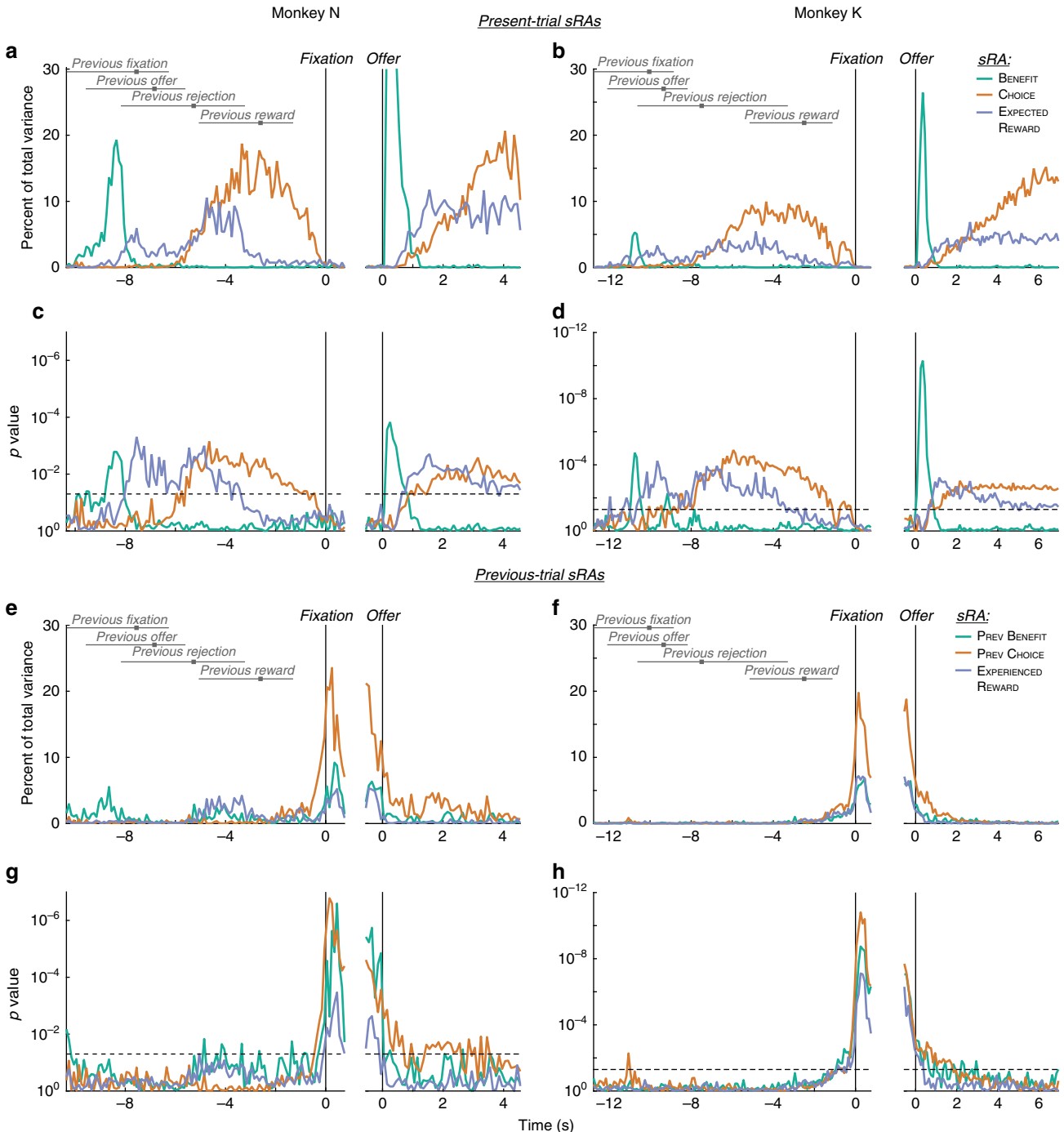

**Fig. 6 Cross-trial representations of task variables. a**, **b** The relevant signal variance (RSV) for the present-trial sRAs of BENEFIT (green), CHOICE (orange), and EXPECTED REWARD (blue) was computed with respect to the previous-trial (left panels) or present-trial (right panels) variables and plotted as a function of time from fixation (left panels) or offer onset (right panels) on the present trial for monkeys N (**a**) and K (**b**). Peak RSV for BENEFIT (**a**, right panel) is not shown so as to conserve the vertical scale across axes. Horizontal gray bars and solid gray squares indicate the 2.5-to-97.5 percentile range and median, respectively, of previous-trial events, as labeled. RSV during the previous trial (left panels) was smaller than during the present trial (right panels) likely due to the wide temporal variability between trials—reward delivery of 0–3 s depending on reward number and ITI of 1–2 s depending on when animal initiated fixation—leading to temporal smearing of otherwise robust, temporally aligned responses. **c**, **d** Log$_{10}$ probability (one-sided, uncorrected) of data in **a** and **b**, respectively, was derived empirically in comparison to random dimensions (see "Methods"). Note: Offer-aligned axes (right panels) of **a–d** recapitulate Fig. 4c–f and are provided for reference. **e**, **f** The RSV for the previous-trial sRAs of PREVIOUS BENEFIT (green), PREVIOUS CHOICE (orange), and EXPERIENCED REWARD (blue) was computed with respect to the previous-trial variables for both temporal alignments (unlike for **a**, **b**, where offer-aligned RSV was with respect to present-trial variables). Other conventions as in **a** and **b**. **g**, **h** Log$_{10}$ probability (one-sided, uncorrected) of data in **e** and **f**, respectively, was derived empirically in comparison to random dimensions (see "Methods"). Horizontal dashed lines in **c**, **d**, **g**, and **h** correspond to $p = 0.05$. As for the present-trial variables, individual units were selective for multiple previous-trial variables (i.e., mixed), and previous-trial sRAs were specific to their targeted variable and largely separable (Supplementary Figs. 26–29 and Supplementary Table 4).

re-evaluated and reversed after initiating the behavior. In contrast, in conventional value-based tasks that require a brief, all-or-nothing response, all deliberation occurs before initiating the behavior, which in turn is executed abruptly and irreversibly, akin to choosing amongst candies in a box of chocolates. Such tasks do not encourage (nor designed to observe) re-evaluation of the initial decision and thus may not elicit the sustained neural representations of value to support re-evaluation.

The distinct demands of abrupt vs. sustained decision tasks may explain the difficulty in validating the widespread model that OFC integrates external value information with subjective preferences to drive decisions[5,9,15,16,46,47,55]. In particular, a key prediction—that variation in OFC's response to an offer accounts for variability in choice (i.e., choice probability, CP)—has not been reported. Unlike prior studies with highly consistent choices[9,56], our animals accepted intermediate offers inconsistently, and thus we were well-poised to assess CP in OFC. Nonetheless, we did not observe choice predictive responses aligned to the offer.

One possible explanation is that value representations in OFC may not drive the initial behavioral policy, but rather inform whether to continue or reverse the initial policy midstream. This putative role of OFC in re-evaluation would not be observed in conventional tasks with brief, all-or-nothing responses. Indeed, optogenetic inactivation of OFC in the rat had no effect on choices in a classical value-based task[57,58]. However, when the same rats were asked to weigh reward against sustained effort, OFC inactivation disrupted normal value-based behavior.

In our sustained-effort task, animals reversed their decision mid-trial at rates proportional to the expected reward. During this period, EXPECTED REWARD maintained the representation of value to inform this re-evaluation, but the serial nature of our recordings precluded linking trial-to-trial variability in EXPECTED REWARD activity (aligned to the offer or rejection) to choices. Going forward, sustained-effort tasks paired with simultaneous recordings and/or temporally targeted manipulations of OFC activity would test the role of OFC in re-evaluating choices mid-execution.

Despite their differences, sustained-effort tasks engage the same fundamental processes—valuation, comparison, and (at least initial) commitment to a behavioral policy—as conventional choice tasks[24,25]. In particular, the representations of BENEFIT, EXPECTED REWARD, and CHOICE were directly analogous to the response categories of Offer Value, Chosen Value, and Taste, respectively, observed in now-classic studies of OFC[15]. Critically, both conventional and sustained-effort tasks require subjects to integrate external value with (potentially changing) internal preferences, goals, and resources—a flexibility untested by traditional studies of self-control or response inhibition[59,60]. For instance, while sustained fixation may have required self-control, our animals applied it adaptively to high-value more than low-value offers. Moreover, goals may vary across tasks. As in our study, agents may maximize reward per unit cost, not only per unit time. For instance, one may reject an offer of $1 to hold a heavy suitcase for one minute simply because the small reward is not worth the high cost. One may even begin holding the suitcase, but reject the cost as too onerous after 30 s. One is not maximizing absolute income, but is conserving resources, which are almost certainly finite and must be allocated judiciously. We believe sustained-effort tasks offer an essential bridge between conventional choice tasks and more ethological behaviors that may depend preferentially on OFC representations[22].

Separately, we sought to address the growing need for a common set of measurements and accompanying statistical framework as more studies analyze high-dimensional neural data[21]. We offer a systematic, principled and statistically rigorous roadmap for population analysis that can be applied, either cohesively

or modularly, to any neural population. Our approach not only de-mixes low-dimensional representations of task-relevant variables, but, critically, emphasizes the relationships between representations, both between different variables at the same time point (i.e., separability) and between different times (i.e., stability). To our knowledge, our paper is the first to assimilate these aspects of population coding—dimensionality reduction, separability, and stability—into a single statistical framework. Moreover, we are the first to apply contemporary population analyses of any sort to OFC.

Unlike other de-mixing techniques, oTDR synthesized several assumptions—regression, orthogonalization, weighting by observation count, and noise reduction—in a single objective function, thereby discovering the optimal linear representations of the variables given the model assumptions. An earlier method applied subsets of these assumptions, but did so serially and thus approximated a solution to the original objective[18]. An alternative method could not accommodate unbalanced designs in which not all combinations of task variables are observed[19], as frequently encountered in decision-making studies. Moreover, oTDR can generalize to any arbitrary number of variables and epochs, including finding sRAs for the same variable in multiple epochs and assuming orthogonalization between any subset of sRAs (see "Methods"; Supplementary Fig. 12). As such, oTDR is a general-purpose method for discovering static representations of known relevant variables in high-dimensional data.

Independently of oTDR, the metrics we developed—including formal definitions of separability and stability, as well as assessments of sensitivity and specificity—are applicable to any set of dimensions, and thus any dimensionality reduction method.

Finally, we introduced novel applications of recent statistical tools[21,36] to test the significance of these metrics (again, applicable to any de-mixing technique). A rigorous statistical framework is crucial to rule-out epiphenomenal findings attributable to the population's intrinsic, low-level features, to which high-dimensional systems are particularly susceptible[21]. Our methods estimated the contribution of these features more accurately—and estimated statistical significance more conservatively—than prior approaches. In particular, random dimensions distributed isotropically overestimate the significance of the high-variance dimensions[12,19], and typical shuffling procedures—randomizing either across time or conditions—account for either the data's dimensionality or temporal smoothness, respectively, but not both.

The present toolkit may be useful to other high-dimensional data sets, such as large-scale simultaneous recordings[26,27,31] or surveys across multiple brain areas where resolving the relative contributions and temporal sequence of representations has been difficult at the single-neuron level[9,61].

## Methods

**Cost-benefit decision-making task.** Two adult male macaque monkeys, N (*Macacca fascicularis*) and K (*M. mulatta*), served as subjects in this study. Prior to experimental use, each animal was prepared surgically with a head-holding device consisting of either a plastic cylinder embedded in acrylic (monkey N) or a titanium post secured directly to the skull (monkey K). During training and while engaged in experiments, daily fluid intake was restricted to maintain adequate levels of motivation; food was freely available. All surgical, behavioral, and animal care procedures complied with National Institutes of Health guidelines and were approved by the Stanford Institutional Animal Care and Use Committee.

Animals were seated in a primate chair in a sound-insulated and dimly lit chamber at a viewing distance of 43 cm (monkey N) or 55 cm (monkey K) from a 20″ CRT computer monitor (ViewSonic G22fb, Walnut, CA) displaying 800 × 600 pixels at 96 Hz. Head position was stabilized using the head-holding device. Eye position was monitored at 1000 Hz with an infrared video tracker (EyeLink 1000, SR Research, Ontario, Canada) mounted on the primate chair with custom hardware; the real-time eye position signal was calibrated periodically[62]. Behavioral control and stimulus presentation were managed by Apple Macintosh G5-based computers (Cupertino, CA) running Expo software written by Peter Lennie

(University of Rochester, NY) with modifications by Julian Brown (Stanford University, CA). Behavioral and stimulus event data were acquired by the Plexon MAP System (Dallas, TX), whereas digital eye position samples were recorded natively on the EyeLink system.

We trained the animals on a novel cost-benefit decision-making task (Fig. 1a). A trial began with the appearance of a fixation point (FP; white annulus, inner/outer diameter 0.3°/0.6°) against a dim background. The animal acquired the FP by directing its gaze within an invisible, circular fixation window around the FP (radius 3° or 1.8°, monkey N or K, respectively). If the animal failed to acquire the FP within 2 s, the FP was extinguished followed by a 1 s delay before reappearance of the FP. The animal was required to maintain fixation for a brief, variable period of time (fixation period; 0.5 - 1 s, uniformly distributed). Fixation breaks during the fixation period terminated the sequence (which was not scored as a trial in future analyses), and the task entered an inter-trial interval (ITI; see below).

Following the fixation period, we presented the offer to the animal (offer period; 0.5 s) as a set of 0, 1, 2, 4 or 8 square icons (0.5° × 0.5°), evenly spaced along an invisible circular ring (6° radius) centered on the FP. The number and color of the icons indicated the offer size, or number of drops of juice the animal would receive as reward for accepting the offer, which varied pseudorandomly from trial to trial, with the constraint that all 5 offer sizes were presented twice every 10 trials. Most offers were presented with yellow icons, each of which represented a single drop of juice. However, to control for the correlation between visual stimulus properties and offer size, half of the 8-reward offers were presented with 8 yellow icons (non-singleton offer) and half with a single purple icon (singleton offer) for half or all experiments with monkey N or K, respectively. The animals learned the association between icons and rewards during training that preceded the current experiments. To control for particular orientations of the icons having a disproportionate effect on behavior or neural responses, we randomly rotated the array of icons around the invisible ring from trial-to-trial, always maintaining an equal angle between icons for a given offer. To provide a temporal cue for 0-reward offers (in which no icons were presented), the center of the FP was filled-in at the onset of the offer period for all offer sizes. (For approximately half of experiments for monkey N, the FP remained an annulus through the offer period, filling-in at the onset of the work period. The delayed FP transition did not affect the timing of responses; Supplementary Fig. 3.)

The offer period was followed by the work period. The offer icons were extinguished and the animal was required to maintain fixation for a constant period of time (4 or 6.5 s for monkey N or K, respectively) that was selected during training such that the animal rarely to maximally accepted the smallest to largest offers, respectively. If the animal maintained fixation until the end of the work period (i.e., accepted the offer), we extinguished the FP and delivered sequentially the drops of juice offered (reward period), with 0.4 s between each drop. (Drop size was regulated by the opening and closing of an electronic solenoid valve, which was calibrated regularly to achieve a constant drop size.) To reject the offer, the animal simply averted its gaze any time during the offer or work periods. The trial then entered a timeout period whose duration was equal to that had the animal accepted the offer. During this timeout period, the screen went blank, and the animal was free to move its eyes, but no reward was delivered.

Following the period when rewards were or would have been delivered, the ITI was imposed. To prevent the animal from reflexively fixating continuously across a string of sequential trials, we required the animal to break fixation at some point during or after the current trial, i.e., any time during the offer or work periods (while the FP was present) or during the reward or ITI periods (during which the FP was absent, but the invisible fixation window was maintained). The ITI period was repeated until the required break occurred. In practice, the animals made numerous saccades during the ITI, and thus this task feature was not engaged outside of initial training. Following the ITI, the FP was presented for the next trial.

We defined the behavioral conditions as the unique combinations of five standard offers (i.e., 0, 1, 2, 4, or 8 rewards) and two choices (i.e., accept or reject the offer), plus the two choices in response to the 8-reward singleton offer, resulting in 12 possible conditions, though some conditions were more frequent than others (e.g., the animal rarely rejected 8-reward offers and rarely accepted 0-reward offers).

### Analysis of behavioral choice

All analyses described here and below were performed with custom scripts written in MATLAB (Mathworks, Natick, MA).

We modeled binary choice behavior to accept or reject the offer on trial $t$ as the logistic function:

$$P(\text{accept}) = \frac{\delta}{1 + e^{-Z}}, \tag{1}$$

where $0 \le \delta \le 1$ specified the maximal accept rate, or saturation point of the psychometric curve. This parameter has previously been used to model the lapse rate, or intrinsic failure rate, of behavior[63]. The exponent $Z$ took the form:

$$Z = \beta_0 + \beta_1(\text{benefit})^\gamma, \tag{2}$$

where $\beta_0$ was a constant and $\beta_1$ determined the influence of the offered benefit [0, 1, 2, 4, 8] on trial $t$ raised to $\gamma$, an exponent typical in economic models to implement a non-linear utility function[64] that was either fixed ($\gamma = 1$) or allowed to vary as a free parameter. The benefit predictor was scaled to the range [0, 1].

Maximum likelihood estimates for the free parameters $\beta_0$, $\beta_1$, $\delta$, and $\gamma$ were obtained independently for each experimental session (i.e., set of trials collected at a given recording site). Maximization was performed by the MATLAB function *fmincon* with constraint $0 \le \delta \le 1$. We considered choices in the normal stimulus condition (in which the number of yellow icons indicated the number of rewards) to represent most closely the animal's cost-benefit function, and therefore excluded the singleton condition (in which 1 purple icon indicated 8 rewards) from all model fits. We took the similarity between the model estimate for the 8-reward normal stimulus and the behavior observed on singleton trials as validation that the animals indeed had learned the value of the singleton stimulus (Fig. 1b).

We solved for two variations of the model, *a* or *b*, with $\gamma$ as a free parameter or with $\gamma = 1$, respectively, obtaining the likelihood $L$ of the data given the model. For each experimental session, we computed the log likelihood ratio (LR) for models *a* and *b*:

$$LR = 2(\log L_a - \log L_b), \tag{3}$$

which, under the null hypothesis that the two models were equally likely, was $\chi^2$ distributed with $df_a - df_b$ degrees of freedom, where df was the model's number of free parameters. Finally, we computed the probability $p(LR)$ of falsely rejecting the alternative hypothesis that the empirical cumulative distribution function of LR across sessions was right-shifted relative to the predicted $\chi^2$ cumulative distribution (i.e., $CDF_{LR}(x) < CDF_{\chi^2}(x)$ for all experimental sessions $x$) by the Kolmogorov-Smirnov test. We rejected the null hypothesis and selected the more complex model *a* to serve for all sessions when $p(LR)$ was <0.05. For monkey N or K, LR = 0.53 or 0.63 and $p(LR) = 0.36$ or 0.037, respectively. Thus, we included $\gamma$ as a free parameter for monkey K only and fixed $\gamma = 1$ for monkey N.

### Electrophysiological recording of OFC

In preparation for physiological recordings, each animal underwent anatomical magnetic resonance imaging (MRI) of the brain. During imaging, a rigid, MRI-visible fiducial marker was attached to the animal's head and could be repositioned identically during subsequent surgery. Using the anatomical images, we identified ideal placement of the recording cylinder that accessed OFC in stereotactic planes. Each animal then underwent a surgical procedure in which a craniotomy was placed and a plastic recording cylinder (Crist Instrument, Hagerstown, MD) was positioned according to the MRI-guided coordinates and relative to the extracranial fiducial marker. The cylinders were centered at stereotaxic coordinates of 36.1 or 35.6 mm anterior to the interaural line, 5.2 or 8.9 mm left of midline, and angled 10.5° or 0.0° anterior to the coronal plane for monkey N or K, respectively. (Note that the reported cylinder location was based on intra-surgical stereotaxic measurements, while the medial-lateral position of individual recording sites, reported in Supplementary Fig. 6, was measured from the MR images and accounted for the angle of the electrode trajectory.)

After the procedure, a small, cylindrical recording grid (Crist Instrument; 1 mm spacing between holes) was placed within the chamber and filled with salinized agarose solution that provided contrast for a second anatomical MRI prescribed such that the imaging planes were parallel (for coronal and sagittal sequences) or orthogonal (for axial sequences) to the recording grid holes. Thus, the final set of MR images shared the same planes as our eventual electrode penetrations, allowing us to visualize the electrode trajectory during each experiment and facilitating precise electrode placement within OFC (Supplementary Fig. 4).

On the basis of previously reported value-related responses[15,42], we concentrated our physiological recordings around the medial orbitofrontal sulcus (mOFS), including the medial and lateral banks and fundus, corresponding to Brodmann areas 11 and 13[65]. We employed standard methods to record the discharge of single units and ensembles of multiple units using extracellular tungsten microelectrodes (FHC Inc., Bowdoin, ME). For each experiment, we advanced the electrode with precision motors that were calibrated to provide a precise (<10 μm) estimate of electrode position (NAN Instruments, Nazareth, Israel), while simultaneously advancing a virtual electrode in the co-registered MR images (Supplementary Fig. 4). We found the correspondence between virtual and actual electrode position to be highly consistent (within 100 μm), as corroborated by the gray-white matter transitions. After waiting 10–20 min for the electrode to stabilize just superior to the mOFS, we slowly advanced the electrode while the animal performed the cost-benefit task, increasing the likelihood that relevant neurons would be active during the selection process. We selected for recording sites where we could isolate at least one single-unit waveform and otherwise applied no additional criteria (taking "all comers") so as to collect as representative a data set of medial OFC as possible (26 or 86 sites, monkey N or K, respectively).

Once selecting a site for recording, we captured and stored all amplified waveforms—discrete excerpts of the continuous voltage time series—that exceeded a predetermined voltage threshold (set such that the event rate in white matter was 1–5 Hz) as digitized samples using the MAP data acquisition system (Plexon Inc., Dallas, TX), which simultaneously captured behavioral events relayed by the Expo software.

### Offline data processing and selection

All analyses were performed on waveforms sorted offline using specialized software (Offline Sorter, Plexon, Inc.) into single units (i.e., single neurons) and multi-units to which we collectively referred as units. The term individual unit referred to either one single unit or one multi-

unit. We identified single units as waveforms whose morphology was stereotypical of single units, consistent over time, and easily distinguishable from other con-current waveforms, and whose timing was separated by at least 1 ms (i.e., outside the minimum refractory period) and was at most weakly correlated with the timing of other waveforms (i.e., unlikely resulting from multiple threshold crossings from a single polyphasic waveform), as done typically in the literature[66]. The remaining, multi-unit waveforms were presumed to originate from two or more neurons. Waveforms deemed to be either electrical artifact or time-locked continuations of a previously counted waveform were excluded.

The final data set included 26 and 131 single units and 42 and 211 multi-units from monkeys N and K, respectively. As reflected in the above counts, we excluded units that were recorded for fewer than 60 trials total (44 and 33 units, monkey N and K, respectively), had fewer than 5 trials in any of the included conditions (i.e., up to 10 unique combinations of offer size and choice; 71 and 133 units), or for which the accompanying behavior was grossly aberrant (monkey K only: 1 session with accept rate of ~55% for 0-reward offers and 1 session with ~10% accept rate for all offers). When a unit was not present during a trial (i.e., mean spike rate averaged across trial of <0.1 Hz for 5 or more consecutive trials, we excluded those trials for that unit; this excluded both units with very low firing rates, as well as trials during which the unit may have been lost to recording. We computed the trial-average firing rate as the mean firing rate across trials in 100 ms time bins for each condition. Because our analysis would ultimately normalize a unit's firing rate by its variability, we excluded 23 units (monkey K only) with extremely low variability (s.d. < 0.5 Hz) as measured across conditions and time bins so that imprecision in estimating variability (to which low-variability units were particularly susceptible) would not result in spurious over-weighting of these units. (Coincidentally, the mean firing rate of the low-variability units was also low, ranging from 0.072 to 0.76 Hz and with a mean of 0.27 Hz compared to 29 Hz for the included population for monkey K.) Finally, because our analysis required that all included conditions were represented by all units, we excluded conditions for which <40% of units met the trial count threshold (i.e., 5 trials/condition), including "0-reward, accept" for both animals and additionally "8-reward, reject" for monkey K.

To compute the trial-average response across $N$ units, $C$ conditions, and $T$ times, we extracted the firing rate $R_n(r, t)$ for each unit $n$ on trial $r$ in the $t^{th}$ non-overlapping, 100 ms time bin aligned to the time of the offer. We then computed the mean firing rate $\tilde{R}_n(c, t)$, or trial-average response, across the trials of condition $c$ (see next paragraph for condition definitions). To standardize the neural response, $\tilde{R}_n$ was $z$-transformed to $\overline{\overline{R}}_n$ such that, for each unit, the mean and standard deviation were 0 and 1, respectively, across all times. Unless otherwise noted, we subtracted the mean response across conditions at each time $t$ (i.e., common-condition response, $CC_n(t)$) from $\overline{\overline{R}}_n(c, t)$, giving $\overline{R}_n(c, t)$. Finally, we compiled the responses $\overline{\overline{R}}_n(c, t)$ and $\overline{R}_n(c, t)$ across units into tensors $\overline{\overline{\mathbf{R}}}$ and $\overline{\mathbf{R}}$, respectively, with dimensions $N \times C \times T$. As such, tensors $\overline{\overline{\mathbf{R}}}$ and $\overline{\mathbf{R}}$ contained the standardized population response, with the responses in tensor $\overline{\mathbf{R}}$ also being mean-subtracted.

We computed two sets of trial-average responses depending on whether the conditions were defined by the present or previous trial. Present-trial responses were computed by sorting trials according to the present trial's condition (i.e., 5 present offers × 2 present choices = 10 possible present-trial conditions; Figs. 1–5 and right panels of Fig. 6a–d), while previous-trial responses were computed based on the previous trial's condition (i.e., 5 previous offers × 2 previous choices; left panels of Fig. 6a–d and all panels of Fig. 6e, f). (Note: the 8-reward singleton condition was excluded when computing population-level dimensions, see below; however, its trial-average response was computed and normalized identically as for the other conditions and then projected onto the dimensions for certain analyses, e.g., Fig. 4b.) To understand the need for the two sorting schemes, consider that, because trials were presented in random order, present-trial responses to any single condition contained all 10 previous-trial conditions (and vice versa for present-trial responses conditionalized on any single previous-trial condition).

**Optimal targeted dimensionality reduction.** To discover low-dimensional representations of the task-relevant variables, we developed optimal targeted dimensionality reduction (oTDR), an extension of the earlier TDR technique[18]. The oTDR method discovered linear combinations of neurons, or low-dimensional representations, that linearly encoded the task-relevant variables, were targeted to variables identified a priori as relevant to the behavioral task.

Formally, we assumed that for a given trial $r$ and time $t$, each task-relevant variable $k$ contributed to the firing rate linearly with coefficient $\beta_{k,n}(t)$, which was summarized in the following single-trial regression model:

$$R_n(r, t) = \beta_{0,n}(t) + \beta_{1,n}(t)P_1(r) + \beta_{2,n}(t)P_2(r) + \beta_{3,n}(t)P_3(r) + \varepsilon, \quad (4)$$

where $R_n(r, t)$ was the firing rate of unit $n$, $\varepsilon$ was independent Gaussian noise, and the predictors were the task-relevant variables: benefit ($P_1$: encoded as {0, 1, 2, 4, 8}), choice ($P_2$: encoded as {0, 1} for rejects and accepts, respectively), and expected reward ($P_3$: given by benefit × choice). By compiling the regression coefficients across $N$ units into the vector $\beta_k(t) \in \mathbb{R}^N$, we defined an axis in the neural population space. We termed this axis a regression axis (RA), as it was obtained from the above regression model. The projection of the population response onto

this vector corresponded to the activity of the RA, as could be read out by a downstream circuit.

In practice, we operated on the trial-average, mean-subtracted, $z$-normalized population response captured in tensor $\overline{\mathbf{R}}$ (see above). We also normalized each task-relevant variable ({$P_1$, $P_2$, $P_3$}) to the range [0, 1]. When computing previous-trial representations, $P_k(r)$ was replaced with the variable's value from the previous trial, i.e., $P_k(r - 1)$, thereby defining the previous-trial variables: previous benefit, previous choice, and experienced reward.

We used the above model to compute two classes of RAs: (1) a single set of static RAs (sRAs) that represented the task-relevant variables across all times in the trial (i.e., for each variable $k$, $\beta_k(t)$ was constrained to be the same $\forall t \in \{1, \dots, T\}$); and (2) a set of dynamic RAs (dRAs) computed independently at each time in the trial to measure how the representations changed across the trial.

**Static low-dimensional representations.** In the first application of oTDR, we sought a single set of sRAs to serve across all times in the trial. We reasoned that a static set of dimensions would both provide a more compact visualization of the data and suggest a means by which downstream neurons could read out distinct task-relevant representations via a static set of weights. Of course, if the dimensions representing the task-relevant variables changed markedly over the trial, then a set of static dimensions would fail to capture some portion of the available signal, which we quantified explicitly (Supplementary Fig. 15).

Toward discovering a static set of RAs, we made three modifications to Eq. (4). First, we replaced the time-varying firing rate with the mean response taken a subset of time bins. While this potentially limited the generalizability of the resulting representation, we reasoned that the sRA should capture the available signal during the epoch when the variable was most behaviorally relevant, and we would assess generalizability subsequently as an empirical question.

In the present study, benefit information was relevant during the 0.5 s of the offer period, during which the animal encoded the incoming sensory information. After the offer period, the valuation and decision process proceeded without external information and thus depended on internal representations of the task variables (namely choice and expected reward), which we computed during the post-offer, work period. In addition to these a priori hypotheses, our selection of the temporal epochs was supported by the dRA analysis (Fig. 5), which showed stable representations of benefit during the offer period and of choice and expected reward during the post-offer period.

Following the above rationale, we segregated the neural data into two epochs defined a priori by the sets of time bins $T_1 \in [0,\dots,0.5 \text{ s}]$ and $T_2 \in (0.5,\dots,4.5 \text{ s}]$ (monkey N) or $(0.5,\dots,7 \text{ s}]$ (monkey K), and, within an epoch, averaged across the time bins to produce the $N \times C$ response matrices $\overline{R}^{(T_1)}$ and $\overline{R}^{(T_2)}$, respectively. We computed the sRA for benefit using $\overline{R}^{(T_1)}$ and computed the sRAs for choice and expected reward using $\overline{R}^{(T_2)}$. (As discussed below, the method supports an arbitrary number of epochs, and the representation(s) of a given variable can be computed in one or more epochs.)

Note that by computing sRAs for choice and expected reward simultaneously in the same epoch, the representations competed to explain shared variance. This competitive process was critical when regressing the same neural responses (i.e., $\overline{R}^{(T_2)}$) onto partially correlated variables, as was the case for choice and expected reward. The sRA for benefit was computed using a distinct set of neural responses (i.e., $\overline{R}^{(T_1)}$) and thus did not compete for shared variance with the other representations. However, because we required all sRAs to be orthogonal (see below), the discovery of any sRA depended on the other sRAs, regardless of the temporal epoch in which it was computed.

In separate analyses, we tested alternative hypotheses about the relevant epochs in which the variables were represented by varying the duration of $T_1$ from 0.5 to 2 s (and shortening $T_2$ accordingly). We also tested the case of having no a priori hypothesis by discovering all three sRAs within a single epoch spanning the entire trial. In all cases, the results did not change qualitatively (Supplementary Fig. 12).

When discovering the previous-trial representations, we defined a single temporal epoch $T_3$ that included the 0.5 s prior to fixation on trial $r$, a time when the animal was under behavioral control but had not yet experienced the offer on trial $r$. Unlike for the present-trial sRAs, we discovered all three previous-trial sRAs (which presented the variables on trial $r - 1$) in the corresponding responses $\overline{R}^{(T_3)}$. All other steps were identical to those in discovering the present-trial sRAs.

Second, the core regression assumption of Eq. (4) operated at the single-trial level and thus had the advantage of weighting each condition by the corresponding number of observations (i.e., trials), which differed systematically across conditions (e.g., animals rarely accepted 1-reward offers and usually accepted 8-reward offers). Therefore and conveniently, conditions for which our estimate of the true, long-run neural response was more reliable (i.e., we had more observations) exerted greater influence on the regression coefficients. However, the trial-average responses, which were needed to combine responses across serially recorded units, did not incorporate the number of observations per condition. Thus, the reliability of the trial-average firing-rate estimates differed across conditions. To control for variable trial counts across conditions, we found that the single-trial model (Eq. (4)) was equivalent to minimizing the Mahalanobis distance of the trial-averaged problem scaled by the square root of trial count (Supplementary Note 1), and thus applied this correction when estimating the coefficients (see matrix M in Eq. (5)).

Third, Eq. (4) placed no constraints on the relationship between the sRAs. However, we would ultimately project the neural responses onto each sRA. For the projections to reflect independent portions of the total neural variance—and, per the goal of oTDR, reflect independent readouts of the task-relevant variables—we required orthogonality between the sRAs. As such, we constrained the regression such that all pairs of $\beta_k$ were orthogonal[67]. We removed the orthogonality constraint when characterizing the intrinsic representation of a given variable or the relationships between representations (e.g., Fig. 3). The constant term $\beta_0$ was never included in the orthogonality constraint (see Supplementary Fig. 17 for relationship between dimensions encoding the common-condition response and the task-relevant variables).

In summary, oTDR discovered static, low-dimensional representations (i.e., sRAs) of the task-relevant variables with the following assumptions:

1. At the level of the single trial, the neural response linearly represented task-relevant variable $k$ as a single dimension $\beta_k$ according to the individual-unit model:

$$R_n(r, T_e) = \beta_{0,n} + \sum_k \beta_{k,n} P_k(r) + \varepsilon,$$

where $R_n(r, T_e)$ was the mean firing rate of unit $n$ on trial $r$ over the temporal epoch $e$ given by set of time bins $T_e$ and $P_k(r)$ was the value of task-relevant variable $k$ on trial $r$.

2. Reliability of the trial-average and time-average response $\bar{R}_n(c, T_e)$ of unit $n$ over all trials of condition $c$ and time bins $T_e$ depended on the number of trials observed for condition $c$.

3. (Optional) Representations $\beta_k$ were orthogonal across all $K$ variables and epochs $e$, i.e., $\beta_1 \perp \ldots \perp \beta_K$.

We incorporated all model assumptions into a single objective function, solved using established optimization tools[67,68]:

$$B = \underset{(B \in \mathbb{R}^{N \times K}) \cup (B_0 \in \mathbb{R}^{N \times 2})}{\operatorname{argmin}} \left[ \left\| S^{(T_1)} \right\|_F^2 + \left\| S^{(T_2)} \right\|_F^2 \right], \quad (5)$$

where

$$S^{(T_1)} = \left( \bar{R}^{(T_1)} - \beta_0^{(T_1)} 1_C^\top - \beta_1 P_1^\top \right) \odot \sqrt{M};$$

$$S^{(T_2)} = \left( \bar{R}^{(T_2)} - \beta_0^{(T_2)} 1_C^\top - \sum_{k=2}^K (\beta_k P_k^\top) \right) \odot \sqrt{M};$$

$$B = \{\beta_1, \ldots, \beta_K\} \in \mathbb{R}^{N \times K};$$

$$B_0 = \left\{ \beta_0^{(T_1)}, \beta_0^{(T_2)} \right\} \in \mathbb{R}^{N \times 2},$$

such that $\beta_1 \perp \ldots \perp \beta_K$ (orthogonality assumption); where $\|\cdot\|_F$ was the Frobenius norm; $\odot$ indicated element-wise matrix multiplication; $\bar{R}^{(T_1)}$ and $\bar{R}^{(T_2)}$ were $N \times C$ matrices specifying the average neural response of unit $n$ in condition $c$ over the first and second temporal epochs (given by time bins $T_1$ and $T_2$), respectively; M was an $N \times C$ matrix specifying the number of trials observed for each unit $n$ and condition $c$; $\beta_0^{(T_1)}$ and $\beta_0^{(T_2)}$ were $N \times 1$ vectors specifying the constant term for the first and second temporal epochs, respectively; $1_C$ was a $C \times 1$ vector of ones; $\beta_k$ was an $N \times 1$ vector specifying the regression coefficient of all units as pertaining to the variable $k$; and $P_k$ was a $C \times 1$ vector specifying the values of variable $k$ across conditions, where $k \in \{1, 2, 3\}$ pertained to the variables benefit, choice, and expected reward, respectively. The columns of B were then normalized to unit vectors, resulting in the final $K$ sRAs.

We designed oTDR to be a general-purpose algorithm for discovering low-dimensional representations of an arbitrary number of $K$ variables and $E$ epochs, while computing dimensions for each variable in one or more epochs and assuming orthogonalization between any subset of dimensions. For simplicity, we stated the objective function above (Eq. (5)) in its narrow form as applied to the current data set. Here we restate the objective in a general form that can be applied to any data set.

$$B = \underset{\left(B^{(T_1)} \in \mathbb{R}^{N \times K^{(T_1)}}\right) \cup \cdots \cup \left(B^{(T_E)} \in \mathbb{R}^{N \times K^{(T_E)}}\right) \cup \left(B_0 \in \mathbb{R}^{N \times E}\right)}{\operatorname{argmin}} \sum_{e=1}^E \left\| S^{(T_e)} \right\|_F^2, \quad (6)$$

where

$$S^{(T_e)} = \left( \bar{R}^{(T_e)} - \beta_0^{(T_e)} 1_C^\top - \sum_k^{K^{(T_e)}} \left( \beta_k^{(T_e)} P_k^{(T_e)\top} \right) \right) \odot \sqrt{M};$$

$$B^{(T_e)} = \left\{ \beta_k^{(T_e)}, \ldots, \beta_{K^{(T_e)}}^{(T_e)} \right\};$$

$$B = B^{(T_1)} \cup \cdots \cup B^{(T_E)};$$

$$B_0 = \left\{ \beta_0^{(T_1)}, \ldots, \beta_0^{(T_E)} \right\},$$

such that all or a subset of columns of B may be constrained to be orthogonal and all columns of B may be projected onto the top $D$ principal components ($D \le N$) of the data (computed by reshaping the tensor $\bar{R} \in \mathbb{R}^{N \times C \times T}$ of data spanning the entire duration of the trial to the 2-D matrix $\bar{R}^{TC} \in \mathbb{R}^{N \times TC}$ for which the principal components were computed; $N$: neurons; $C$: conditions; $T$: time bins); where $\bar{R}^{(T_e)}$ was an $N \times C$ matrix specifying the average neural

response of unit $n$ in condition $c$ in temporal epoch $T_e$; $\beta_0^{(T_e)}$ was an $N \times 1$ vector specifying the constant term for epoch $T_e$; $P_k^{(T_e)}$ was a $C \times 1$ vector specifying the values of task-relevant variable $k$ assumed to be represented in epoch $T_e$; and $\beta_k^{(T_e)}$ was an $N \times 1$ vector specifying the corresponding regression coefficient. Note in general, for a given epoch $T_e$, all or a subset of task-relevant variables $\left\{ P_1^{(T_e)}, \ldots, P_{K^{(T_e)}}^{(T_e)} \right\}$ and the corresponding regression vectors $\left\{ \beta_1^{(T_e)}, \ldots, \beta_{K^{(T_e)}}^{(T_e)} \right\}$ may be present (i.e., $K^{(T_e)} \le K \; \forall e \in \{1, \ldots, E\}$). Furthermore, a given task-relevant variable and its corresponding regression vector may be shared across multiple epochs (i.e., the number of columns of matrix B is given by $\sum_{e=1}^E K^{(T_e)}$). The columns of B were normalized to unit vectors, resulting in the final sRAs. The remaining terms and conventions are defined above.

In Supplementary Fig. 12e, f, we applied the generalized objective function to the current data set to show how representations of a single task-relevant variable can be computed in multiple temporal epochs and how orthogonalization can be applied to a subset of representations. We also discuss the rationale for these assumptions.

As stated above, the objective function could accommodate an additional de-noising assumption such that the task-relevant representations were limited to the high-variance subspace, i.e., space spanned by the top $D$ principal components (PCs). We did not apply the de-noising assumption in the present study, in part because the noise-reducing effect of averaging over multiple time bins $T_e$ obviated the need for additional noise reduction. More importantly, limiting the data to the high-variance subspace would compromise the subsequent hypothesis testing. Specifically, as described below, we tested the significance of a given representation (i.e., sRA) by comparing it to random vectors biased to the space occupied by the data. By limiting the data to the high-variance subspace, the random vectors would also be limited to this high-variance subspace—a subspace much smaller than that spanned by the original data. As a result, the random vectors would underestimate the full space of possible representations, and our estimates of the probability of obtaining a given sRA by chance, as computed from the random vectors, would be exaggerated.

The present oTDR technique extended the previous TDR method[18]. Both techniques were "targeted" in that variables of interest were defined a priori and then dimensions representing those variables were discovered in the high-dimensional neural data. However, the previous technique used separate, ad-hoc algorithms to apply each assumption serially: computing a set of linear representations $\beta_k$ of variable $k$ via Eq. (4), projecting $\beta_k$ onto the top PCs (also see ref. [12]), and then orthogonalizing the set of $\beta_k$ using a greedy algorithm. Each step distorted $\beta_k$ from the original vectors, and thus the final vectors were no longer necessarily as close to linear representations of the targeted variables as possible given the model assumptions. In addition, the prior technique relied on solving for $\beta_k$ at a single moment in time, which compromised the generality of the representation across all relevant times. In contrast, by incorporating all assumptions into a single objective function (Eq. (5)), oTDR satisfied the model assumptions simultaneously and thereby discovered targeted dimensions that were optimal given all model assumptions.

Of note, as for oTDR, the prior technique implicitly weighted conditions by the number of observations (assumption 2) through use of the single-trial model (Eq. (4)). However, use of this model complicated simultaneous application of the orthogonalization and de-noising steps, since application of these steps would require the objective function to solve for each unit serially and transition back-and-forth between single-trial and trial-average data. Our finding that the single-trial model could be recast as a scaled trial-average problem facilitated the simultaneous application of all assumptions.

**Dynamic low-dimensional representations**. Thus far, we applied oTDR to discover static representations, or sRAs, of the task-relevant variables. In our second application, we measured how the representations changed across the trial. To this end, we discovered a time series of dynamic representations, or dRAs, discovered independently in each time bin. The dynamic and static analyses shared the basic assumptions outlined in the single-trial model (Eq. (4)), but the methods differed slightly. In particular, the dynamic analysis operated on shorter time bins, which made the regression coefficients more susceptible to random variation in firing rate. As such, we applied several additional steps to reduce the impact of random variation.

First, prior to solving the regression, we noised-reduced the neural data by eliminating the low-variance dimensions, which contributed less to representations of the task-relevant variables (as confirmed in Supplementary Fig. 18). To identify the low-variance dimensions, we transformed the 3-D tensor $\bar{R}$ (with data spanning the entire trial) to the 2-D matrix $\bar{R}^{TC}$ (dimensions $N \times TC$) and performed principal components analysis (PCA) on $\bar{R}^{TC}$. (Note, by transforming $\bar{R}$ to $\bar{R}^{TC}$, the resulting PCs explained variance related either to time or task condition.) We projected the trial-average, mean-subtracted response $\bar{R}_n(c, t)$ onto the top $D$ PCs, as ordered from those explaining the most to least variance, generating the noise-reduced responses $\breve{R}_n(c, t)$. We selected $D$ as the approximate inflection point of the log variance explained as a function of PC. The subspace

spanned by the top PCs ($D = 8$ or 20) explained 45% or 32% of total variance across time and conditions for monkey N or K, respectively.

Second, we averaged adjacent 100 ms time bins into single, non-overlapping 200 ms time bins that did not straddle $t = 0$. The variable $t$ in the dynamic analyses refers to these wider, 200 ms time bins.

Third, we applied $L_2$ regularization (i.e., ridge regression) to mitigate random variation in the coefficients $\beta_{k,n}(t)$. The regularization penalized large values of $\beta_{k,n}(t)$ and assumed the contribution of a given unit $n$ was distributed across variables $k$. This assumption was supported by the observed mixed selectivity of individual units, i.e., lack of clustering of sRA coefficients along the horizontal and vertical meridians (Fig. 3). Of note, we did not apply $L_2$ regularization when discovering the sRAs (Eq. (5)) for two reasons. First, the noise-reducing effect of regularization was less necessary in the static analysis given the more precise measurements of the neural response, as discussed. Second, in discovering the special case of non-orthogonalized sRAs (e.g., Fig. 3), we were interested in accurately measuring the absolute relationships between task-relevant representations, which would be artificially distorted by assuming a priori (via regularization) that coefficients were distributed across variables. In contrast, we used the dRAs primarily to understand the change in relationships between representations over time (relative to synthetic data to which regularization was also applied; see below), and thus the absolute relationships were less critical. We tested the effects of PCA-based noise reduction, time bin width (100, 200, or 500 ms), and regularization in separate analyses (Supplementary Fig. 22).

Finally, unlike the sRAs, we did not orthogonalize the set of dRAs because we were interested in the relationship between the representations—either of different variables at the same time, or of the same variable at different times—and orthogonalization would have distorted these relationships.

In solving the static regression model (Eq. (5)), numerical matrix optimization was required to include the orthogonality assumption. However, because orthogonalization was not included in the dynamic analysis, the model for the dRAs had a closed-form solution and could be expressed at the individual-unit level:

$$
\begin{aligned}
\breve{R}_n(c,t)\sqrt{M_n(c)} = {} & \beta_{0,n}(t) + \beta_{1,n}(t)P_1(c)\sqrt{M_n(c)} + \beta_{2,n}(t)P_2(c)\sqrt{M_n(c)} \\
& + \beta_{3,n}(t)P_3(c)\sqrt{M_n(c)} + \lambda_n(t)\left\| \left[\beta_{0,n},\beta_{1,n},\beta_{2,n},\beta_{3,n}\right] \right\|_2^2,
\end{aligned}
\tag{7}
$$

where $\breve{R}_n(c,t)$ was the noise-reduced, trial-average response of unit $n$ for condition $c$ at time $t$, $M_n(c)$ was the number of trials observed per condition, $\lambda_n(t)$ was a scalar parameter governing the impact of the regularization term, and the regularization term, $\|\cdot\|_2$, was the Euclidian norm of the vector of the constant term $\beta_{0,n}$ and the three coefficients $\beta_{k,n}$ for the task-relevant variables $P_k \in \{\text{benefit, choice, expected reward}\}$. We found $\lambda_n(t)$ empirically and independently for each unit and time bin via leave-one-out cross-validation on conditions and selecting the value of $\lambda$ that minimized the mean squared error for Eq. (7) on the left-out test condition. For certain units and time bins, the error was minimized by $\lambda = \infty$, implying excessive unexplained variance; in these cases, we set $\beta_{k,n}(t) = 0$ for all $k$. We compiled coefficients $\beta_{k,n}(t)$ across units to generate $N$-dimensional vectors for each variable $k$ that we then normalized to unit vectors, referred to as the "dRA($t$) for [variable $k$]" in the main text.

**Projection of neural response onto static regression axes.** Each sRA defined a linear combination of neurons that represented the targeted task-relevant variable. To read out these representations, we projected the neural responses onto each sRA:

$$
\text{proj}_k(t) = \bar{R}(t)^\top \beta_k,
\tag{8}
$$

where $\bar{R}(t)$ was an $N \times C$ matrix of the population response extracted from tensor $\bar{\mathbf{R}}$ at time $t$, and $\beta_k$ was the $N$-dimensional sRA corresponding to variable $k$. The $C$-dimensional vector $\text{proj}_k(t)$ gave the activity of sRA$_k$ at time $t$ for each of the $C$ conditions.

We limited projection-based analyses to the static regression axes. As discussed, a primary aim of the oTDR static analysis was to separate, or de-mix, the population encoding of the task-relevant variables into independent representations. Because the sRAs were orthogonal, the projection onto a given sRA was the optimal readout of the population response (given the model assumptions) as related linearly to the targeted variable of interest and minimally related to the other task-relevant variables. However, if the RAs were non-orthogonal, as was the case for the dRAs (see above), then the resulting projections necessarily would contain representations of multiple task-relevant variables. While these mixed projections may separate the task-relevant variables more than the mixed selectivity at the individual-unit level, they would not offer a maximally independent readout. Therefore, projections onto the dRAs were of limited utility for interpreting the population activity or for positing how downstream circuits may read out the population activity. For this reason, we did not project the population response onto the dRAs, and consequently did not perform the subsequent variance-based analyses (see below) that depended on these projections.

**Variance explained by static regression axes.** We measured the variance $V_k(t)$ at time $t$ explained by sRA$_k$ (i.e., the sRA targeted to task-relevant variable $k$) as the variance of $\text{proj}_k(t)$ across conditions normalized by the cross-condition variance summed over all dimensions (i.e. units), which we expressed as a percentage:

$$
V_k(t) = \frac{\text{var}(\text{proj}_k(t))}{\sum_n^N \text{var}(\bar{R}_n(:,t))} \times 100,
\tag{9}
$$

where $\text{var}(\bar{R}_n(:,t))$ was the variance of neural response across conditions for unit $n$ at time $t$.

**Relevant and irrelevant signal variance.** Though each sRA was designed to represent a specific variable, some of the variance explained by a given sRA may have been unrelated to this on-target variable. Furthermore, this unrelated variance may have been related to an alternative off-target variable. We were interested in measuring the variance explained by a given sRA$_k$ (targeted to variable $k$) that was related or unrelated to variable $q$, and so developed metrics for relevant or irrelevant signal variance (RSV or ISV, respectively):

$$
\text{RSV}_{k,q}(t) = V_k(t)r_{k,q}^2(t),
\tag{10}
$$

$$
\text{ISV}_{k,q}(t) = 100 - \text{RSV}_{k,q}(t),
\tag{11}
$$

where $V_k(t)$ was the variance explained by sRA$_k$ at time $t$ and $r_{k,q}^2(t)$ was the squared Pearson's correlation coefficient ($0 \le r^2 \le 1$) between $\text{proj}_k(t)$ and the condition-matched values of variable $q$ (note, by convention, the subscript $k$ in $r_{k,q}^2$ did not refer to variable $k$ directly, but rather to the projection onto sRA$_k$). Thus RSV and ISV were in units of variance, lower-bounded by 0, and together summed to $V_k(t)$. By convention, when referring to the on-target variable (i.e., $q = k$), we simplified the subscripted indexing: $\text{RSV}_{k,q=k} = \text{RSV}_k$ and $\text{ISV}_{k,q=k} = \text{ISV}_k$.

When we computed RSV and ISV for the on-target variable (i.e., $q = k$), the metrics provided a useful account of how well the sRA was sensitive and specific, respectively, to the variable of interest. However, when we computed RSV or ISV for off-target variables (i.e., $q \ne k$), the term could misattribute on-target variance as off-target variance when variables $k$ and $q$ were correlated (e.g., the values of benefit [0, 1, 2, 4, 8, 1, 2, 4, 8] and expected reward [0, 0, 0, 0, 0, 1, 2, 4, 8] were correlated, $r = 0.54$).

To consider the effect of correlated variables, let $a$ and $b$ be on- and off-target variables, respectively, moderately correlated according to the Pearson's correlation coefficient $r_{a,b}$. Let $p$ be the projection of the neural population response onto sRA$_a$ designed to detect variance related to $a$. To simplify the formulation, assume $a$, $b$, and $p$ are mean-centered, though this is not essential to the argument. Let $a$ and $p$ be correlated by a relatively large coefficient $r_{p,a}$. Consequently, we would compute a relatively high on-target RSV$_a$. However, given $r_{a,b}$, we would also expect a relatively high correlation $r_{p,b}$ and thus a high off-target RSV$_{a,b}$, giving a false impression that RSV$_a$ is non-specific to variable $a$ and explains substantial variance related to variable $b$. In the extreme case, when $r_{a,b} = 1$, then $r_{p,b} = r_{p,a}$, and thus RSV$_{a,b}$ = RSV$_{a,a}$.

To control for the correlation $r_{a,b}$ between variables, we employed a method known as semi-partial correlation[69] to isolate the relationship between $b$ and $p$ that was not explained by $r_{a,b}$. We replaced the Pearson's correlation coefficient $r_{p,b}$ with the semi-partial correlation coefficient $\rho_{p,b|a}$, as given by

$$
\rho_{p,b|a} = \frac{r_{p,b} - r_{p,a}r_{a,b}}{\sqrt{1 - r_{a,b}^2}}.
\tag{12}
$$

Therefore, to isolate the portion of variance related to off-target variables, while controlling for the correlation between on- and off-target variables, we computed off-target RSV as

$$
\text{RSV}_{k,q}(t) = V_k(t)\rho_{k,q|k}^2(t),
\tag{13}
$$

where $\rho_{k,q|k}^2(t)$ was the squared semi-partial correlation coefficient (Eq. (12)) between $\text{proj}_k(t)$ and the component of off-target variable $q$ not explained by on-target variable $k$. We continued to computed RSV for on-target variables (i.e., $q = k$) per Eq. (10). Thus,

$$
\text{RSV}_{k,q}(t) = \begin{cases} V_k(t)r_{k,q}^2(t), & \text{when } q = k \\ V_k(t)\rho_{k,q|k}^2(t), & \text{when } q \ne k \end{cases}.
\tag{14}
$$

Regarding ISV, on-target ISV$_k$ indicated the amount of variance explained by sRA$_k$ that was not correlated with the on-target variable $k$ and thus was available for correlation with the off-target variables. However, the concept of off-target ISV was of limited utility, because one expected for the variance explained by a given sRA to be unrelated to the off-target variables. Indeed, high on-target RSV indicated high off-target ISV. Therefore, we only computed ISV for the on-target variables.

The metrics RSV and ISV were not limited to sRAs computed by oTDR and could be computed for any arbitrary axis. However, for dimensionality reduction techniques agnostic to the variables of interest (e.g., PCA), the notion of on- and off-target variables was not well-defined. In the case of PCA (Supplementary Fig. 16), the dimensions (i.e., PC$_d$) corresponded to vectors in the $N$-dimensional

space that explained the greatest cross-condition and temporal variance, from $d = 1$ to $N$. As with the sRAs, we projected the neural population response onto each $PC_d$ and measured the variance explained (Eq. (9)). For a given task-relevant variable $k$, we defined the on-target $PC_{d'}$ as that which explained the greatest cumulative $RSV_{d,k}$ with respect to variable $k$ (as per Eq. (10)) across all time bins, and likewise defined $d'$ as the number of the corresponding PC. Subsequently, we recomputed off-target $RSV_{d,k}$ (as per Eq. (13)) for the remaining PCs (i.e., when $d \neq d'$), while maintaining on-target $RSV_{d',k}$ per Eq. (10) when $d = d'$.

**Angle between regression axes**. To measure the similarity between either the static or dynamic representations, we computed the pairwise angle in degrees between RAs $\beta_i$ and $\beta_j$:

$$\theta_{ij} = \cos^{-1}(|\beta_i^\top \beta_j|) \times 180/\pi. \quad (15)$$

Note that in this general formulation, $\beta_i$ and $\beta_j$ were unit vectors and could refer to several types of RA pairs: sRAs for variables $i$ and $j$, dRAs for the same variable at times $i$ and $j$, or dRAs for different variables $i$ and $j$ at the same or different times. Angles of 0° indicated the representations were identical (or perfectly opposite), whereas angles of 90° indicated the representations were maximally unrelated (i.e., orthogonal). We took the absolute magnitude of the dot product so as treat angles equidistant from 90° as equivalent (e.g., angles of 0° and 180° were both coded as $\theta = 0°$), and thus refer to $\theta_{ij}$ as the folded angle. In so doing, we emphasized the absolute similarity of the RA pair (e.g., angles of 0° and 180° both indicated that the same units contributed by the same absolute amount to the pairs of RAs). However, this construction of $\theta_{ij}$ was insensitive to differences in the sign of representation, as when the neuronal contributions to $\beta_i$ and $\beta_j$ remain similar in absolute strength but reverse sign, such as exhibited by the example unit in Fig. 2d. Therefore, exclusively to observe changes in sign, we constructed the unfolded angle $\theta'_{ij}$ as:

$$\theta'_{ij} = \begin{cases} \cos^{-1}(\beta_i^\top \beta_j) \times \frac{180}{\pi}, & \text{when } \beta_i^\top \beta_j < 0 \\ \text{Not-a-number}, & \text{otherwise} \end{cases}, \quad (16)$$

and $\theta'_{ij}$ was therefore limited to the range [90, 180°] (Supplementary Fig. 23).

**Boxcar analysis to define periods of putative stability**. To define and characterize periods of putative stability, we fit boxcar functions to the time course of angles $\theta_{ij}$ for each reference dRA($i$) computed at a fixed time $i$ in comparison to dRA($j$) for all times $j$. Each time course corresponded to a row in the heat maps in Fig. 5c, d and is shown with the boxcar fits in Supplementary Fig. 20. The temporal span of each boxcar indicated the period of putative stability, while boxcar height $\theta_{boxcar}$ indicated the average similarity of the representations during that period (see Supplementary Fig. 21 for all boxcar metrics). Note that boxcars were fit to 90° − $\theta_{ij}$, thus periods of high similarity had large values of $\theta_{boxcar}$ but small values of $\theta_{ij}$.

To test statistical stability, we compared our observations to identical metrics made on synthetic firing rate data from a hypothetical neural population that did not encode the task variables but preserved the correlations between units and across time (see below). In each synthetic data set, we computed the average similarity $\hat{\theta}_{boxcar}$ during each period of putative stability, where periods were defined by the boxcar fits from the veridical data. We defined a period as statistically stable when the observed similarity ($\theta_{boxcar}$) was significantly greater than the null values of $\hat{\theta}_{boxcar}$ compiled across synthetic data sets (i.e., $p(\theta_{boxcar}) < 0.01$; Supplementary Fig. 21c, f).

**Alignment index**. In addition to comparing the similarity between one-dimensional RAs via the angle analysis above, we were interested in assessing the overlap between pairs of subspaces spanned by sets of RAs or other basis vectors. To measure the overlap between subspaces $U_1$ and $U_2$, we used a custom metric, termed the alignment index $A$, adapted from a recent report[36]:

$$A = \frac{\text{Tr}(U_1^\top U_2 U_2^\top U_1)}{\min(D_1, D_2)}, \quad (17)$$

where $U_1$ and $U_2$ were orthogonal matrices of dimensions $N \times D_1$ and $N \times D_2$, respectively, with $D$ basis vectors arranged in columns, and Tr(.) was the matrix trace. The numerator measured the amount of overlap between $U_1$ and $U_2$, and the denominator normalized the index by the minimum dimensionality of the subspaces (i.e., the maximum possible overlap). Thus, the alignment index ranged from 0 (orthogonal) to 1 (completely aligned) and was invariant to the order of $U_1$ and $U_2$.

**Null model using random dimensions in neural space**. A major goal of the present study was to develop and apply generalizable, unbiased tools for significance testing of low-dimensional representations of high-dimensional data. We developed two statistical approaches to contextualize the low-dimensional features by generating control data sets of either (1) random dimensions that captured the inter-neuronal correlations (i.e., dimensionality) of the data, or (2) random

synthetic responses that captured the data's dimensionality and temporal correlations (i.e., temporal smoothness).

To model the correlations between neurons (i.e., dimensionality), we generated random dimensions that reflected the high-dimensional space occupied by the neural data[36]. That is, the density of random dimensions in the high-dimensional space was proportional to the frequency that the neural population occupied a given region of the space.

To provide an intuition for the impact of dimensionality, consider a population of two neurons that tended to fire together. The dimension reflecting co-activation would be more likely to contain data than the dimension reflecting exclusive activation of one neuron or the other. This asymmetry should be reflected in the set of random dimensions intended to model the chance probability of a given dimension (or its properties) given the data. Alternatively, if the set of random vectors were evenly distributed throughout two-dimensional space (as in most analyses), then many random dimensions would be over-represented relative to how frequently the neural population actually occupied those dimensions, thereby overestimating the rarity of the commonly occupied dimensions and generating spuriously small p-values. Instead, by generating random dimensions reflecting the correlational structure between neurons, our estimate of the rarity of a given dimension was less biased and the resulting p-values were more conservative.

The details of the random dimension method are described elsewhere[36]. Briefly, we calculated the covariance matrix $\Sigma$ of the neural responses $\bar{\mathbf{R}}^{TC}$ (dimensions $N \times TC$) across all times and conditions of the task (as was done for computing the PCs above). We then generated random dimension $\hat{v}_i$ aligned to the dimensionality of the neural data as:

$$\hat{v}_i = \text{orth}\left(\frac{U\sqrt{S}\vec{v}_i}{\|U\sqrt{S}\vec{v}_i\|_2}\right) \quad (18)$$

where the eigenvectors of $\Sigma$ were in the columns of the $N \times N$ matrix U, and the corresponding eigenvalues were on the diagonal of the $N \times N$ diagonal matrix S; $\vec{v}_i$ was a random $N$-dimensional vector with each element drawn independently from a normal distribution with zero mean and unity variance; and orth(.) returned the orthonormal basis of an input matrix. We repeated this procedure for $i = 1$–10,000, returning a set of random dimensions that had the specified inter-neuronal covariance structure $\Sigma$. For statistical analysis, we projected the neural data onto the set of random dimensions, generating $\widehat{\text{proj}}_i(t)$ for each random dimension $\hat{v}_i$. Using $\widehat{\text{proj}}_i(t)$, we computed identical metrics as when projecting the data onto a dimension of interest (e.g., when computing variance explained by an sRA). Across random dimensions, this produced a null distribution for a given metric from which we estimated the probability of obtaining a given value of that metric by chance given the inter-neuronal correlational structure of the data.

In general, computing a given metric based on the projection onto a random dimension (i.e., $\widehat{\text{proj}}_i(t)$) was straightforward. However, computing off-target $\widehat{\text{RSV}}_{i,q}$ for random dimension $i$ and off-target variable $q$ required specification of the semi-partial correlation $\hat{\rho}^2_{i,q|k}(t)$ that isolated the portion of $q$ that was independent of the Pearson's correlation $\hat{r}^2_{i,q=k}$ between the specific projection $\widehat{\text{proj}}_i(t)$ and the on-target variable $k$. Therefore, we first computed the on-target correlation $\hat{r}^2_{i,q=k}$ (Eq. (10)) for random dimension $i$ and on-target variable $k$, which we then used to compute the semi-partial correlation $\hat{\rho}^2_{i,q|k}(t)$ (via Eq. (12)) and, in turn, off-target $\widehat{\text{RSV}}_{i,q}$ (via Eq. (13)).

Separately, in the case of computing the similarity between two dimensions (i.e., computing their angle $\theta$, Eq. (15); Supplementary Tables 2–4), we generated the null distribution of angles $\hat{\theta}$ by computing the angle between all pairs of random dimensions $\hat{v}$.

**Null model using neural population control data**. Our analyses of the dynamics of the representations (e.g., dRA stability, Fig. 5) required we account not only for the data's dimensionality, as in the random dimensions technique above, but also for the data's temporal correlations, or smoothness. These two aspects of the population's correlational structure limited how much a representation could change, and therefore could trivially account for similarity in the representations across time. To provide an intuition, correlations between units, or dimensionality, restricted the subspace a representation could occupy (as discussed above). At the limit, if the dimensionality were unity, then a representation could not vary and would be similar, indeed identical, across all times. Separately, correlations in an individual unit's response over time, i.e., temporal smoothness, restricted how quickly a representation could change. If temporal correlations were very high, then even if the dimensionality permitted dissimilar representations, the population may not have been able to transition to an alternative representation within the span of a trial.

We therefore sought to generate surrogate control data that captured both the data's dimensionality and temporal smoothness. However, traditional shuffling procedures—that randomize either across time points or task conditions—maintain either the dimensionality or smoothness of the data, respectively, but not both, and therefore are prone to overestimate the significance of a given finding. Instead, we employed a recent method, neural population control[21], to generate synthetic firing-rate data that preserved both the dimensionality and temporal

                                                                                                ARTICLE

smoothness of the original data but were otherwise random. Briefly, we quantified the covariance of the original data across time and neurons so as to define a joint probability distribution of firing rates that was maximally entropic given the data's dimensionality and temporal smoothness. We then generated surrogate data sets by sampling firing rates from this distribution at each time bin and for each neuron. Given $N$ units in the original data set, we generated 1000 surrogate data sets of $N$ neurons each, with the original data's correlational structure preserved within a given surrogate data set (i.e., individual data sets were conditionally independent from one another). By computing identical metrics against the surrogate data sets as for the original data, we compared a given value observed in the original data to the distribution of surrogate values. Any population feature, such as the stability of a representation, which appeared in the surrogate data sets would be considered epiphenomenal, i.e., an expected byproduct of the data's dimensionality and temporal smoothness. Whereas population features that rarely occurred in the surrogate data sets would be considered statistically significant.

In its original application[21], the method for generating surrogate data sets was used to test the significance of the strength of encoding of a given signal. However, we observed that, by removing the coding properties from the surrogate data sets, we could control for intrinsic spatiotemporal correlation in accounting for the similarity of coding dimensions across time. To our knowledge, this is the first statistical test of stability for low-dimensional representations.

**Separability and reliability of representations**. The present study sought to identify the encoding of task-relevant variables at the level of individual units and separate these signals into distinct low-dimensional population representations. Our efforts therefore depended on whether the population representations were indeed separable, a concept for which we develop a definition here. Though prior studies have shared similar goals[18–20], the concept of separability was not formally defined or tested.

A rigorous examination of separability is imperative when making claims about de-mixing high-dimensional representations since, as we outline below, noisy estimates of the individual representations can contribute to exaggerated estimates of the independence between representations, thus overestimating the very basis for de-mixing. Also at stake are claims of mixed selectivity, since a population of units selective exclusively for variable $A$ can appear to also encode variable $B$ when estimates of the representations of $A$ or $B$ are noisy, and thereby generate the spurious impression that selectivity is mixed.

The concept of separability depends on conventional correlation, but the statistical hypotheses are distinct. Conventional correlation statistics address the chance of falsely concluding orthogonal representations are correlated, whereas we were interested in the opposite extreme: whether two (possibly correlated) representations carried (at least some) independent information that could be separated at the level of the readout. Therefore, as our null hypothesis, we considered the extreme case of two perfected correlated (or anti-correlated) representations (i.e., $|r| = 1$), but the estimates of each had been corrupted by independent noise. These representations would become increasingly less correlated with increasing imprecision in the estimates, and yet would still not convey separable information about their respective variables. In contrast, we defined two representations as separable when their correlation was significantly less than this null-hypothetical correlation.

To develop this definition, let vectors $\beta'_A$ and $\beta'_B$ be the true $N$-dimensional linear representations (i.e., RAs) of task-relevant variables $A$ and $B$ defined for a population of $N$ neurons. As above, $\beta'_A$ and $\beta'_B$ are separable when they contain independent information, i.e., when their correlation coefficient is less than unity, $|r_{A'B'}| < 1$. Likewise, we define the null hypothesis that the representations are not separable, i.e., perfectly correlated, as:

$$H_0 : |r_{A'B'}| = 1.$$

However, we do not have access to the true representations—only to the sample estimates $\beta_A$ and $\beta_B$ and their correlation $|r_{AB}|$, as measured from the data. Following standard hypothesis testing procedure, we calculate the probability $P$ that we obtained a particular value of $|r_{AB}|$ under the null hypothesis, i.e., $P(|r_{AB}||H_0)$. Here we develop a sampling distribution for $|r_{AB}|$ given $H_0$.

The relationship between $|r_{AB}|$ and $|r_{A'B'}|$ is given by Spearman[70], who observed that $|r_{AB}|$ decreases as independent noise, $\varepsilon_A$ and $\varepsilon_B$, is added to the true representations, i.e.:

$$\beta_A = \beta'_A + \varepsilon_A$$
$$\beta_B = \beta'_B + \varepsilon_B. \tag{19}$$

At the limit, the measured correlation between two perfectly correlated variables goes to zero as $\varepsilon \gg \beta'$. We do not directly measure $\varepsilon_A$ and $\varepsilon_B$, but instead measure the effect of independent noise on the reliability of our measurements of $\beta_A$ and $\beta_B$. Our measure of reliability is based on the correlation, $r_{AA}$ or $r_{BB}$, between repeated samples of $\beta_A$ or $\beta_B$, which we simulate via a bootstrap procedure, detailed below. If reliability were perfect, then $r_{AA} = 1$ and $r_{BB} = 1$. Spearman offered the following correction for the attenuating effect of independent noise (i.e., imperfect reliability) on the correlation between two variables[70]:

$$|r_{AB}| = |r_{A'B'}|\sqrt{r_{AA}r_{BB}}. \tag{20}$$

Under the null hypothesis (i.e., $|r_{A'B'}| = 1$) the above expression reduces to:

$$|\hat{r}_{AB}| = \sqrt{r_{AA}r_{BB}} \text{ (given } H_0). \tag{21}$$

(Note use of the "hat" symbol to indicate the null-hypothetical value.)

In detail, the bootstrap procedure to estimate $r_{AA}$ or $r_{BB}$ was as follows. We generated $S$ resampled data sets by randomly selecting $Q_{n,c}$ trials with replacement from each condition $c$ and unit $n$ given $Q_{n,c}$ original trials for that condition and unit, such that the number of trials per condition for a given unit was consistent across the original data and resampled data sets ($S = 700$; chosen for the maximum number of trials per unit). Note that we $z$-normalized the resampled data for a given unit via the single mean and standard deviation observed for that unit in the full data set. This had the effect of decreasing our subsequent reliability measure and making our determination of separability more conservative. The alternative—$z$-normalizing independently within each resampled data set—would minimize trial-to-trial response fluctuations and thereby overestimate reliability and separability.

Within each resampled data set, we used to oTDR without orthogonalization to calculate the three sRAs for the task-relevant variables benefit, choice, and expected reward. Here we discuss the representation $\beta_A$ (i.e., sRA) of variable $A$; an identical procedure was used for the other two variables. We computed all pairwise correlations between $\beta_{A,i}$ and $\beta_{A,j}$, where $i$ and $j$ were different resampled data sets (given $S$ data sets, we calculated $(S^2 - S)/2$ correlations). We compiled these correlations into the distribution $r_{AA}$ (shown in Supplementary Fig. 7a, c).

To estimate $|\hat{r}_{AB}|$ under the null hypothesis, we computed the distribution $|\hat{r}_{AB}|$ (shown in Supplementary Fig. 7b, d) for each pair of variables $A$ and $B$ (given three variables, we computed three pairs) according to Eq. (21) using the just-compiled distributions of reliability $r_{AA}$ and $r_{BB}$:

$$|\hat{\boldsymbol{r}}_{AB}| = \sqrt{\boldsymbol{r}_{AA} \odot \boldsymbol{r}_{BB}},$$

where $\odot$ indicated element-wise multiplication.

Finally, we performed a one-tailed $t$-test to determine the probability $P(|r_{AB}||H_0)$ of falsely rejecting the null hypothesis that the values in $\hat{r}_{AB}$ were from a distribution with mean $|r_{AB}|$ (as observed in the data) in favor of the alternative hypothesis that $\hat{r}_{AB}$ came from a distribution with greater mean (i.e., that the observed $|r_{AB}|$ was less than the null-hypothetical value $|\hat{r}_{AB}|$). When $P$ was sufficiently small, we concluded that the representations $\beta_A$ and $\beta_B$ were separable.

Our definition of separability serves as a useful test for whether two representations contain independent information about their respective variables and highlights the confounding influence of noisy representations (a feature in virtually all empirical data sets) on observing separability erroneously. Despite these advantages, we consider the following limitations. Because we used a bootstrap procedure (i.e., resampled data sets shared some trial-level data), the resulting reliability estimates (i.e., $r_{AA}$ and $r_{BB}$) were admittedly biased upward, thus making our final test of separability less conservative (via Eq. (21)). In addition, because $\beta_A$ and $\beta_B$ were derived from the same set of neural responses and because the corresponding variables $A$ and $B$ may be correlated, we cannot guarantee that the respective noise components, $\varepsilon_A$ and $\varepsilon_B$, were themselves independent, an assumption of Eq. (20). If $\varepsilon_A$ and $\varepsilon_B$ were correlated, then the measured correlation $|r_{AB}|$ would increase without a change in the null-hypothetical $|\hat{r}_{AB}|$ (which depends on $\varepsilon_A$ and $\varepsilon_B$ separately, not on their interaction), thus making our conclusions regarding separability more conservative.

In addition to the separability analysis, we used the resampled data sets to measure the reliability of each unit's contribution to a given representation. As discussed above, the representation $\beta_{A,s}$ of variable $A$ was derived for each resampled data set $s$, and each element $\beta_{A,s}^{(n)}$ specified the contribution of unit $n$ to the representation. We compiled the distribution $\beta_A^{(n)}$ across the $S$ data sets. The standard deviation of the distribution was proportional to the reliability of unit $n$'s contribution (shown by error bars in Fig. 3a, c). Separately, via a two-tailed $z$-test, we computed the probability of falsely rejecting the null hypothesis that the distribution $\beta_A^{(n)}$ included the value of zero. When this $p$-value was sufficiently small, we concluded that unit $n$ significantly encoded variable $A$. Subsequently, we tested whether the population's selectivity was indeed mixed by whether the proportion of units significantly encoding multiple variables was greater than expected by chance (Supplementary Table 1). This test controlled for the confounding effect of noisy estimates of $\beta_A^{(n)}$ and $\beta_B^{(n)}$ appearing as mixed selectivity for variables $A$ and $B$, when in fact one or both estimates may have been indistinguishable from zero, and thus the true selectivity was for only one or neither variable.

**Reporting summary**. Further information on research design is available in the Nature Research Reporting Summary linked to this article.

## Data availability
The source data underlying Figs. 1b, c, 2a–e, 3a–d, 4a–f, 5a–f, and 6a–h are available for download at https://github.com/danielkimmel/Kimmel_NatComm_2020.

## Code availability

Custom MATLAB code for oTDR and the neural population metrics and statistics presented here can be found at https://github.com/danielkimmel/oTDR.

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

## Acknowledgements
We thank Antonio Rangel, Leo Sugrue, and Kelsey Clark for contributions to task design; Valerio Mante for guidance on neural population analysis; David Sussillo, Jonathon Shlens, and Michael Mozer for useful feedback on the manuscript; and Julian Brown, Jessica Powell, Sania Fong, and Jamie Sanders for technical assistance. This work was supported by the Howard Hughes Medical Institute (D.L.K. and W.T.N.); Stanford Bio-X, Leon Levy Foundation, and NIH Grants T32 GM007365, R25 MH086466, and T32 MH015144 (D.L.K); Columbia Neurobiology and Behavior Program (G.F.E.); and Sloan Foundation, McKnight Foundation, Simons Foundation Collaboration on the Global Brain, NSF Neuronex, and Gatsby Charitable Foundation (J.P.C.).

## Author contributions
Conceptualization and design, D.L.K. and W.T.N.; data acquisition, D.L.K.; formal analysis, D.L.K., G.F.E., and J.P.C.; writing, D.L.K.; review and editing, D.L.K., G.F.E., J.P.C., and W.T.N.

## Competing interests
The authors declare no competing interests.
