## [Peer Review File · Nature Communications]

Reviewers' Comments:

Reviewer #1:

Remarks to the Author:

The authors have addressed all of my concerns. The editor also asked me to respond to Reviewer 3. My response is here as well:

Overall, this does not change my recommendation, which is positive. However, I think the reviewer raises some good points. To wit:

I think that reviewer 3 and the authors are talking at cross purposes to some extent. The reviewer argues in the section "This concern is reinforced..." that fixation is essentially zero effort. The authors believe that this does require some effort. However, I think the reviewer is likely correct that fixation is a particularly degenerate form of effort, very different from the authors' example of climbing stairs. NOT moving is in many ways the canonical opposite of putting effort into something. This is not, in my view, a fatal concern - but is nonetheless a moderate sized limitation that ought to be acknowledged in a more forthcoming manner.

Section on "There is a clear 'neuro'" - I see what the authors are saying here, but I think they are being uncharitable to the reviewer. The original paper, as all three reviewers pointed out, buries its lede. As the reviewer indicates, it took him/her "significant study" to identify the take-home. That was true for me as well. The revised paper improves on this, but I think the authors would be well-advised to spend a little more time whittling down the key parts of the paper - Abstract, Intro, first paragraph of Discussion - into strong compelling narratives. The authors' defensiveness to the reviewers may be convincing in getting the paper published, but there's an opportunity for the paper to be more influential if the authors take this valid criticism seriously.

I will also note, to the authors, that including all revised text in the reply to reviewers document, which in my experience as a reviewer is more common than not, would greatly help the reviewer in assessing the changes to the text, without having to flip back and forth between documents.

Reviewer #2:

Remarks to the Author:

The authors present a revised manuscript following review of the previous version. These changes are largely positive, and have made several key improvements, including clarifying the specific contribution of their statistical method, refining the specific claims about OFC function supported by their results and providing more extensive connections to and discussion of other literature and results on both OFC in value-guided decision making and dimensionality-reduction methods and techniques. I appreciate the work the authors have undertaken and I think the result is an improved manuscript that communicates both a set of intriguing results regarding neural representations in a brain area that is the focus of much interest in the domain of value-based choice as well as an important advance in methods available for the statistical assessment of representations at the level of neural populations.

Overall, I think the findings specific to OFC representations in value-guided choice and the statistical method for neural population analysis together will be of interest to a relatively large subset of the field (though not necessarily overlapping subsets!). The main neuroscience finding -- of stable representations that reflect key task and decision variables specifically during temporal epochs in which they are behaviorally relevant -- is now clearly communicated, and the relationship between this finding (enabled in a rigorous fashion by their novel statistical method) and the particular features of their behavioral task is now much more fully reported and discussed.

Specifically, the presentation of the statistical method now includes a more general form and it is more transparent there and elsewhere in the text how it would generalize to other tasks and settings. In addition, the description of the boxcar analysis is now clearer and the motivation more transparent. Amendments to the figures are very helpful for the reader and overall the reworked introduction and discussion are now clearer and more compelling.

One question remains - is there a reason the authors would not include the new analysis of activity in the choice sRA for early/late rejection provided in their letter? While a temporally resolved, trial-by-trial analysis is not possible in this task and data acquisition setting, I think this analysis goes some way to relating the temporal evolution of choice-reflective activity to the presumed dynamics of 'when' the choice (accept/reject) is made (even though it is not necessarily a punctate event). While I do not think it need be in the main results (especially as attention is drawn to the relation between choice activity and the median rejection time), it is informative and useful to the interested reader. As the lack of relation between the timing of accept/reject decisions in behavior and the timing of neural representations regarding benefit, choice and expectation is one of the weaknesses of the study (that is not a criticism, just that here as in other experiments, the design precludes certain approaches), it goes some way to addressing an overarching concern across all reviewers about the relation between the results and the task.

Regarding issues raised by other reviewers:

- A more explicit statement from the authors regarding the relation of the current study to other dimensionality reduction techniques and their own work is totally warranted, and I think the authors here provide this with an explicit discussion that makes the contribution of the present work more clear, and states that the statistical framework is applicable to projections determined using other techniques.
- Further, the authors have made the relation of the present findings to previous findings and current understanding of OFC function more clear, from the introduction through to the discussion.
- Lastly, I had some trouble following the reviewer's concern about the time-course of broken fixation and the rationality or otherwise of rejecting an offer for reward that requires an effortful action to obtain -- it is well established that motivation to perform actions to obtain rewards fluctuates even in a deprived state, and all manner of costs are integrated into goal-directed decisions, as well as evaluation of other alternatively rewarding actions/activities (grooming, resting etc). That said, the new text that more clearly discusses the relation between classic rapid decision making and more temporally extended effortful decisions is a valuable contribution to the manuscript and will certainly be useful for some readers.

Response to Reviewers

Nature Communications (NCOMMS-20-05432-T)

Replies are shown in blue and preceded with “>>>”.

Reviewer #1 (Remarks to the Author):

The authors have addressed all of my concerns. The editor also asked me to respond to Reviewer 3. My response is here as well:

>>> We are grateful to the reviewer for all his/her help improving our manuscript and supporting its publication.

Overall, this does not change my recommendation, which is positive. However, I think the reviewer raises some good points. To wit:

I think that reviewer 3 and the authors are talking at cross purposes to some extent. The reviewer argues in the section "This concern is reinforced..." that fixation is essentially zero effort. The authors believe that this does require some effort. However, I think the reviewer is likely correct that fixation is a particularly degenerate form of effort, very different from the authors' example of climbing stairs. NOT moving is in many ways the canonical opposite of putting effort into something. This is not, in my view, a fatal concern - but is nonetheless a moderate sized limitation that ought to be acknowledged in a more forthcoming manner.

>>> We appreciate the reviewer helping resolve our confusion with Reviewer 3's remarks. We agree that, at face value, fixation may seem like a trivial task. But in fact decades' of studies have demonstrated the effortful nature of sustained fixation. Most directly, Blanchard, Hayden, et al. (2014, 2015) and Cai, Lee, et al. (2009) used sustained fixation to operationalize economic cost in value-based decision tasks. We now underscore this point and cite these references when describing the task (Results):

To accept an offer, the animal maintained visual fixation for a constant duration (work period)—an effortful process with economic cost³⁵⁻³⁷—and then received the promised reward.

In addition, in the oculomotor literature, numerous papers from Krauzlis, Keller, Wurtz, and Munoz, among others, argue that fixation is an active process by showing sustained activity in regions of the superior colliculus responsible for generating saccadic eye movements, leading to a prominent theory that what appears as static fixation is actually a series of microsaccades of near-zero amplitude, blurring the distinction between “moving” and “not moving.” Finally, early psychophysical work by Yarbus (1967) demonstrated that (human) primates make frequent spontaneous saccades (~3 Hz) and rarely engage in sustained fixation for more than ~300 ms. Thus tasks requiring fixation of 4 - 6 seconds require subjects to significantly alter their innate behavior.

We appreciate the limitations of the “climbing stairs” analogy and have now updated the relevant passage with an analogy that more faithfully captures the nature of effort in the task (Discussion):

As in our study, agents may maximize reward per unit cost, not only per unit time. For instance, one may reject an offer of \$1 to hold a heavy suitcase for one minute simply because the small reward is not worth the high cost. One may even begin holding the suitcase, but reject the cost as too onerous after 30 seconds. One is not maximizing absolute income, but is conserving resources, which are almost certainly finite and must be allocated judiciously.

Section on "There is a clear 'neuro'" - I see what the authors are saying here, but I think they are being uncharitable to the reviewer. The original paper, as all three reviewers pointed out, buries its lede. As the reviewer indicates, it took him/her "significant study" to identify the take-home. That was true for me as well. The revised paper improves on this, but I think the authors would be well-advised to spend a little more time whittling down the key parts of the paper - Abstract, Intro, first paragraph of Discussion - into strong compelling narratives. The authors' defensiveness to the reviewers may be convincing in getting the paper published, but there's an opportunity for the paper to be more influential if the authors take this valid criticism seriously.

>>> We are very grateful to all the reviewers not only for the time and care they took reviewing the manuscript, but especially for suggesting we highlight the paper's key contributions and take-home message. We took these suggestions to heart and made major revisions to the Intro and Discussion in the prior revision. We greatly apologize to the reviewers for any perceived defensiveness we may have inadvertently communicated. Quite the contrary, we recognize that our original draft was too dense and "buried its lede", sacrificing the clarity and impact of our message. We believe the reviewers' suggestions substantially improved the quality of the manuscript.

Our difference with R3's comments were exclusively with the applicability of the results and methods, which we and the other reviewers felt were of "broad interest." It is possible that R3's difference of opinion was entirely due to the difficulty in identifying the "take-home" message in the original version. Since it appears that this message is clear in the revision, we hope this also resolves R3's concerns.

Furthermore, we appreciate R1's advice that we whittle down the revised manuscript even further. We have simplified the prose throughout the current manuscript, including the Abstract, Intro, and Discussion, and have shortened the main sections from 8200 to 6000 words. We have done so primarily by removing redundant description of results that were already conveyed more efficiently in the figures and/or tables (while referring the reader to these sources) and, in some cases, moving certain details to figure legends or supplementary figures. As such, we have preserved the major findings and critical discussion points, including those requested by the reviewers in the prior revision.

I will also note, to the authors, that including all revised text in the reply to reviewers document, which in my experience as a reviewer is more common than not, would greatly help the reviewer in assessing the changes to the text, without having to flip back and forth between documents.

>>> We thank the reviewer for the excellent suggestion, which we have followed for this revision. We apologize for the inconvenience caused by our original reply.

Reviewer #2 (Remarks to the Author):

The authors present a revised manuscript following review of the previous version. These changes are largely positive, and have made several key improvements, including clarifying the specific contribution of their statistical method, refining the specific claims about OFC function supported by their results and providing more extensive connections to and discussion of other literature and results on both OFC in value-guided decision making and dimensionality-reduction methods and techniques. I appreciate the work the authors have undertaken and I think the result is an improved manuscript that communicates both a set of intriguing results regarding neural representations in a brain area that is the focus of much interest in the domain of value-based choice as well as an important advance in methods available for the statistical assessment of representations at the level of neural populations.

Overall, I think the findings specific to OFC representations in value-guided choice and the statistical method for neural population analysis together will be of interest to a relatively large subset of the field (though not necessarily overlapping subsets!). The main neuroscience finding -- of stable representations that reflect key task and decision variables specifically during temporal epochs in which they are behaviorally relevant -- is now clearly communicated, and the relationship between this finding (enabled in a rigorous fashion by their novel statistical method) and the particular features of their behavioral task is now much more fully reported and discussed.

Specifically, the presentation of the statistical method now includes a more general form and it is more transparent there and elsewhere in the text how it would generalize to other tasks and settings. In addition, the description of the boxcar analysis is now clearer and the motivation more transparent. Amendments to the figures are very helpful for the reader and overall the reworked introduction and discussion and now clearer and more compelling.

>>> We are deeply grateful to the reviewer for the extensive time and attention s/he has given to our manuscript, not to mention the extremely helpful suggestions, which have improved both the rigor of the analyses and clarity of the message.

One question remains - is there a reason the authors would not include the new analysis of activity in the choice sRA for early/late rejection provided in their letter? While a temporally resolved, trial-by-trial analysis is not possible in this task and data acquisition setting, I think this analysis goes some way to relating the temporal evolution of choice-reflective activity to the presumed dynamics of 'when' the choice (accept/reject) is made (even though it is not necessarily a punctate event). While I do not think it need be in the main results (especially as attention is drawn to the relation between choice activity and the median rejection time), it is informative and useful to the interested reader. As the lack of relation between the timing of accept/reject decisions in behavior and the timing of neural representations regarding benefit, choice and expectation is one of the weaknesses of the study (that is not a criticism, just that here as in other experiments, the design precludes certain approaches), it goes some way to addressing an overarching concern across all reviewers about the relation between the results and the task.

>>> We thank the reviewer for this helpful suggestion and how have included the analysis in Supplementary Figure 9 with appropriate reference in the main text (Results, "Reading out population activity..."):

To more directly test the link between sRA dynamics and choice timing, we compared CHOICE activity for early vs. late rejections and found that choice selectivity emerged later on late-rejection trials, consistent with the representation reflecting the underlying decision dynamics (Supplementary Figure 9).

Regarding issues raised by other reviewers:

- A more explicit statement from the authors regarding the relation of the current study to other dimensionality reduction techniques and their own work is totally warranted, and I think the authors here provide this with an explicit discussion that makes the contribution of the present work more clear, and states that the statistical framework is applicable to projections determined using other techniques.

- Further, the authors have made the relation of the present findings to previous findings and current understanding of OFC function more clear, from the introduction through to the discussion.

- Lastly, I had some trouble following the reviewer's concern about the time-course of broken fixation and the rationality or otherwise of rejecting an offer for reward that requires an effortful action to obtain -- it is well established that motivation to perform actions to obtain rewards fluctuates even in a deprived state, and all manner of costs are integrated into goal-directed decisions, as well as evaluation of other alternatively rewarding actions/activities (grooming, resting etc). That said, the new text that more clearly discusses the relation between classic rapid decision making and more temporally extended effortful decisions is a valuable contribution to the manuscript and will certainly be useful for some readers.

>>> We appreciate this summary of R3's initial critiques and are glad that our revision appears to have addressed them sufficiently. We share R2's uncertainty regarding R3's specific concerns around the rationality of rejecting an offer and share R2's intuition. Nonetheless, we have revised the manuscript to reference prior literature in which economic cost is operationalized as sustained fixation (Results):

To accept an offer, the animal maintained visual fixation for a constant duration (work period)—an effortful process with economic cost³⁵⁻³⁷—and then received the promised reward.